# Creation of a long-acting nanoformulated dolutegravir

Brady Sillman[1], Aditya N. Bade[1], Prasanta K. Dash[1], Biju Bhargavan[1], Ted Kocher[1], Saumi Mathews[1], Hang Su[1], Georgette D. Kanmogne[1], Larisa Y. Poluektova[1], Santhi Gorantla[1], JoEllyn McMillan[1], Nagsen Gautam[2], Yazen Alnouti[2], Benson Edagwa[1] & Howard E. Gendelman[1,2]

Potent antiretroviral activities and a barrier to viral resistance characterize the human immunodeficiency virus type one (HIV-1) integrase strand transfer inhibitor dolutegravir (DTG). Herein, a long-acting parenteral DTG was created through chemical modification to improve treatment outcomes. A hydrophobic and lipophilic modified DTG prodrug is encapsulated into poloxamer nanoformulations (NMDTG) and characterized by size, shape, polydispersity, and stability. Retained intracytoplasmic NMDTG particles release drug from macrophages and attenuate viral replication and spread of virus to CD4+ T cells. Pharmacokinetic tests in Balb/cJ mice show blood DTG levels at, or above, its inhibitory concentration$_{90}$ of 64 ng/mL for 56 days, and tissue DTG levels for 28 days. NMDTG protects humanized mice from parenteral challenge of the HIV-1$_{ADA}$ strain for two weeks. These results are a first step towards producing a long-acting DTG for human use by affecting drug apparent half-life, cell and tissue drug penetration, and antiretroviral potency.

[1] Department of Pharmacology and Experimental Neuroscience, University of Nebraska Medical Center, Omaha, NE 68198, USA. [2] Department of Pharmaceutical Sciences, University of Nebraska Medical Center, Omaha, NE 68198, USA. Correspondence and requests for materials should be addressed to B.E. (email: benson.edagwa@unmc.edu) or to H.E.G. (email: hegendel@unmc.edu)

Antiretroviral therapy (ART) has changed what was once a life-endangering human immunodeficiency virus type one (HIV-1) infection to a chronic manageable disease. Rapid immune suppression, opportunistic infections, and malignancies were attenuated by antiretroviral drug (ARV) therapy[1,2]. Patients adhering to defined ART regimens can lead a fully productive life, experience limited infection-related morbidities and prevent what was once a rapid inevitable death[3]. However, treatment advancements have come at some cost. These include toxicities, adherence failures, costly regimens, and common viral mutations linked to specific resistance patterns[4]. A means to combat each can be achieved through drug regimen adherence facilitated by long-acting slow effective release antiretroviral therapy (LASER ART)[5] as defined by slow drug dissolution, enhanced lipophilicity, improved bioavailability, and limited off-target toxicities. Such changes in drug formulation affect the frequency of drug

**Fig. 1** Synthesis and characterization of MDTG. **a** A fourteen-carbon fatty-acid modified DTG prodrug (MDTG) was synthesized creating hydrophobic crystals at a final drug yield of 82.8%. **b** Absorption bands at 2915 cm$^{-1}$ and 2850 cm$^{-1}$ in the MDTG Fourier-transformation infrared spectrum (FTIR) illustrate the methyl C–H asymmetric and symmetric stretching of the myristic acid alkyl group. Bands at 1795 cm$^{-1}$ in the myristoyl chloride and 1750 cm$^{-1}$ in the MDTG FTIR spectrum correspond to carbonyl C = O stretching of the myristic acid acyl halide that reacts as the ester is formed in MDTG. **c** X-ray diffraction (XRD) analysis for DTG and MDTG demonstrates the crystalline structures of both drugs. **d** Aqueous solubility of DTG and MDTG demonstrates the decreased solubility of MDTG. ****$P < 0.0001$ DTG vs. MDTG. **e** IC$_{50}$ was determined in vitro in MDM by HIV-1 RT inhibition after DTG and MDTG treatment over a range of concentrations (0.01–1000 nM). Chemical modification of DTG did not affect antiretroviral drug activity (56.7 nM and 62.5 nM for DTG and MDTG, respectively; $P = 0.8397$). Results are shown as the mean ± SEM of three replicates. Results from **d** were analyzed by two-tailed Student's $t$ test ($n = 10$ DTG, 12 MDTG; $t = 20.1$, degrees of freedom = 20). Results from **e** were analyzed by nonlinear regression least squares fit

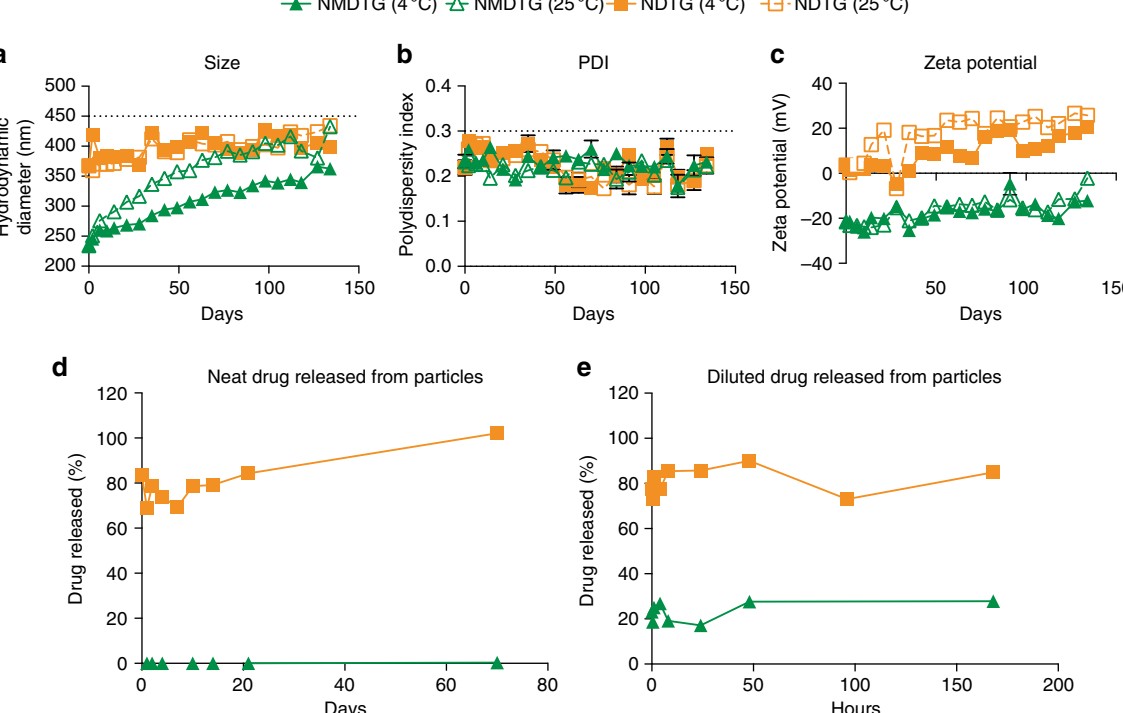

**Fig. 2** Nanoformulation stability and release kinetics. Nanoformulations were synthesized by high-pressure homogenization using poloxamer 407 (P407) as the excipient for DTG and MDTG. **a–c** Formulation stability (up to 134 days) was measured by (**a**) particle hydrodynamic diameter (size), (**b**) polydispersity index (PDI), and (**c**) zeta potential as determined by dynamic light scattering (DLS). NDTG and NMDTG stability were tested at both 4 °C and 25 °C. **d, e** Nanoparticle release kinetics were assessed by drug released from the nanoparticles during storage, in (**d**) freshly manufactured (neat) formulation and (**e**) formulation diluted for in vivo administration. All results are shown as the mean ± SEM of at least three replicates

administration. Reductions in disease co-morbidities follow the maintenance of effective antiretroviral drug concentrations in blood, body fluids, and viral tissue reservoir for months or longer[6]. The end result is improved treatment outcomes.

While such a directive is attractive, not all ARVs can easily be transformed into long-acting medicines. Drug solubility, dissolution, metabolism, protein-binding, and excretion rates are not uniform amongst each drug; and each influence the ARV's half-life and biodistribution profile[7,8]. Long-acting medicines depend upon the maintenance of plasma drug concentrations, which are linked to depot formation and dissociation within the reticuloendothelial system. Such changes in drug depot formation and dissociation, referred to by the term apparent half-life, distinguish it from the drug's intrinsic half-life. Whether a medicine can be found with significant antiretroviral efficacy, limited resistance patterns and high tolerability for conversion into a long-acting compound also remain uncertain. While ARVs have been developed with long-acting properties based on their solubility and protein-binding capacities, none are currently used clinically and only a few (long-acting injectable cabotegravir and rilpivirine) are in clinical trials[6]. For these, the drug formulations require larger injection volumes, which can lead to injection site reactions. Adjusted dosing intervals are also required when inter-patient pharmacokinetic (PK) profiles come operative[9–11].

To these ends, our laboratory developed a process to transform standard daily or twice-daily ARVs into hydrophobic and lipophilic drug nanocrystals to extend the apparent half-life by altering drug solubility and metabolic patterns[12,13]. We developed chemical modification and polymer coating techniques to convert native ARV's into LASER ART. Herein, DTG, an approved second-generation integrase strand transfer inhibitor (INSTI) with potent activity against HIV-1 holds a protein-adjusted inhibitory concentration$_{90}$ (PA-IC$_{90}$) of 64 ng/mL and a high

barrier to resistance[14]. As such, DTG is unique amongst other compounds by its robust resistance profile and measured efficacy in inhibiting HIV-1 growth.

In the current study, alteration of the DTG chemical structure is made through myristoylation of the native compound to create a water-insoluble prodrug, termed MDTG, with commensurate crystal formation. When the drug crystals are packaged into nanoparticles, MDTG is rapidly taken up by human monocyte-derived macrophages (MDM) residing for prolonged periods inside the cells. Drug is slowly released from the particle. MDTG undergoes rapid bioconversion to its parent compound in the presence of esterases contained in biological fluids. This process yields a pharmacologically active product[15]. Such chemical and biological outcomes improve antiretroviral activities up to 30-fold. PK and pharmacodynamic (PD) profiles in mice are also significantly improved over a native drug formulation, exhibiting 5.3-fold extension in drug apparent half-life, broad tissue distribution, and increased antiretroviral efficacy. Our data provide evidence that DTG conversion into a long-acting, slow release formulation is readily achievable with reductions in dose and dosing intervals that could extend to one month or longer. As such, the drug-encased nanoparticles could be employed as a first-step measure to improve regimen adherence, limit adverse reactions by maximizing drug loading and reducing excipient usage. Such improved treatment measures can also minimize drug fatigue and facilitate drug penetrance into viral reservoirs.

## Results

**Synthesis and characterization of MDTG.** A fourteen-carbon fatty-acid modified DTG prodrug (MDTG) was synthesized by esterification of the 7-hydroxyl group of DTG with myristic acid with a final yield of 82.8% (Fig. 1a). The chemical structure of

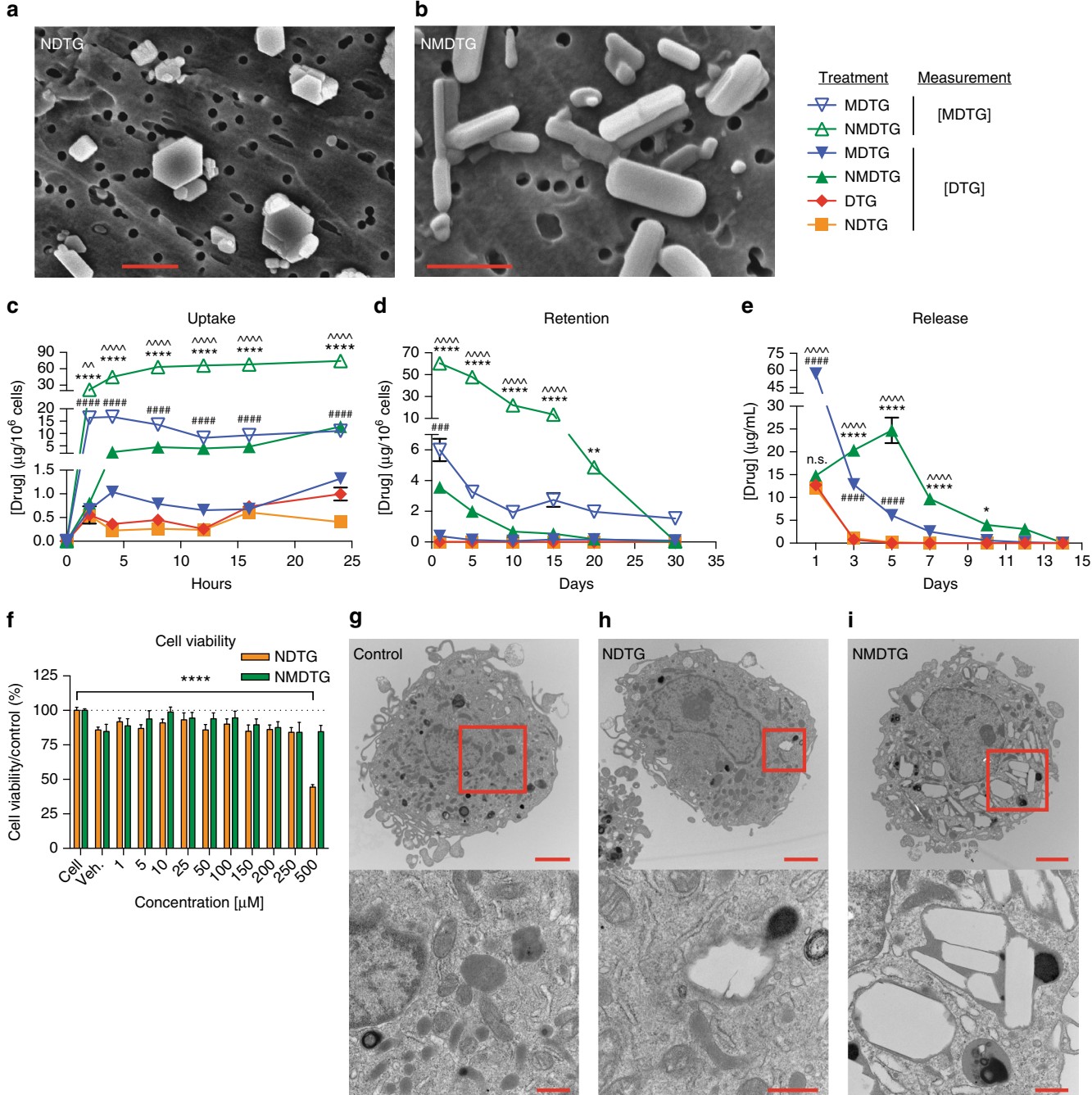

**Fig. 3** Nanoparticle characterization. **a**, **b** Particle morphology was assessed by scanning electron microscopy (SEM). **a** Note that NDTG particles are of heterogeneous size and shape, and show both cuboidal and rod-shaped morphologies. **b** NMDTG particles are more uniform, with dominant rod-shaped morphologies (scale bars = 1 μm). **c** Drug uptake in MDM was measured over a 24-h period with equal drug concentrations (100 μM). Uptake of NMDTG was at or more than a half-log greater than its nonmodified control. ****$P < 0.0001$ NMDTG vs. NDTG. ####$P < 0.0001$ MDTG vs. DTG. ^^$P = 0.0054$, ^^^^$P < 0.0001$ NMDTG vs. MDTG. **d** Drug retention in MDM was measured over a 30-day observation period demonstrating a log greater retention of the NMDTG compared to the nonmodified control. **$P = 0.0048$, ****$P < 0.0001$ NMDTG vs. NDTG. ###$P = 0.0004$ MDTG vs. DTG. ^^^^$P < 0.0001$ NMDTG vs. MDTG. **e** Drug release from MDM was measured over a 14-day observation period demonstrating slowed and prolonged release from NMDTG-treated cells through 10 days. *$P = 0.0156$, ****$P < 0.0001$ NMDTG vs. NDTG. ####$P < 0.0001$ MDTG vs. DTG. ^^^^$P < 0.0001$ NMDTG vs. MDTG. **f** Cell viability was assessed in MDM by MTT assay 24 h after NDTG or NMDTG treatment over a range of concentrations (1–500 μM). Results were normalized to untreated control cells. ****$P < 0.0001$ 500 μM NDTG vs. control (untreated) cells. **g–i** Transmission electron microscopy (TEM) of (**g**) control, (**h**) NDTG, and (**i**) NMDTG loaded MDM after 8-h drug treatment. Note the paucity of particles in the NDTG-treated cells compared to the NMDTG-treated MDM (scale bars = 2 μm, upper panels; 500 nm, lower panels). Results are shown as the mean ± SEM of three biological replicates. Results from **c**, **d**, **e**, **f** were analyzed by two-way ANOVA with Bonferroni's multiple comparison tests

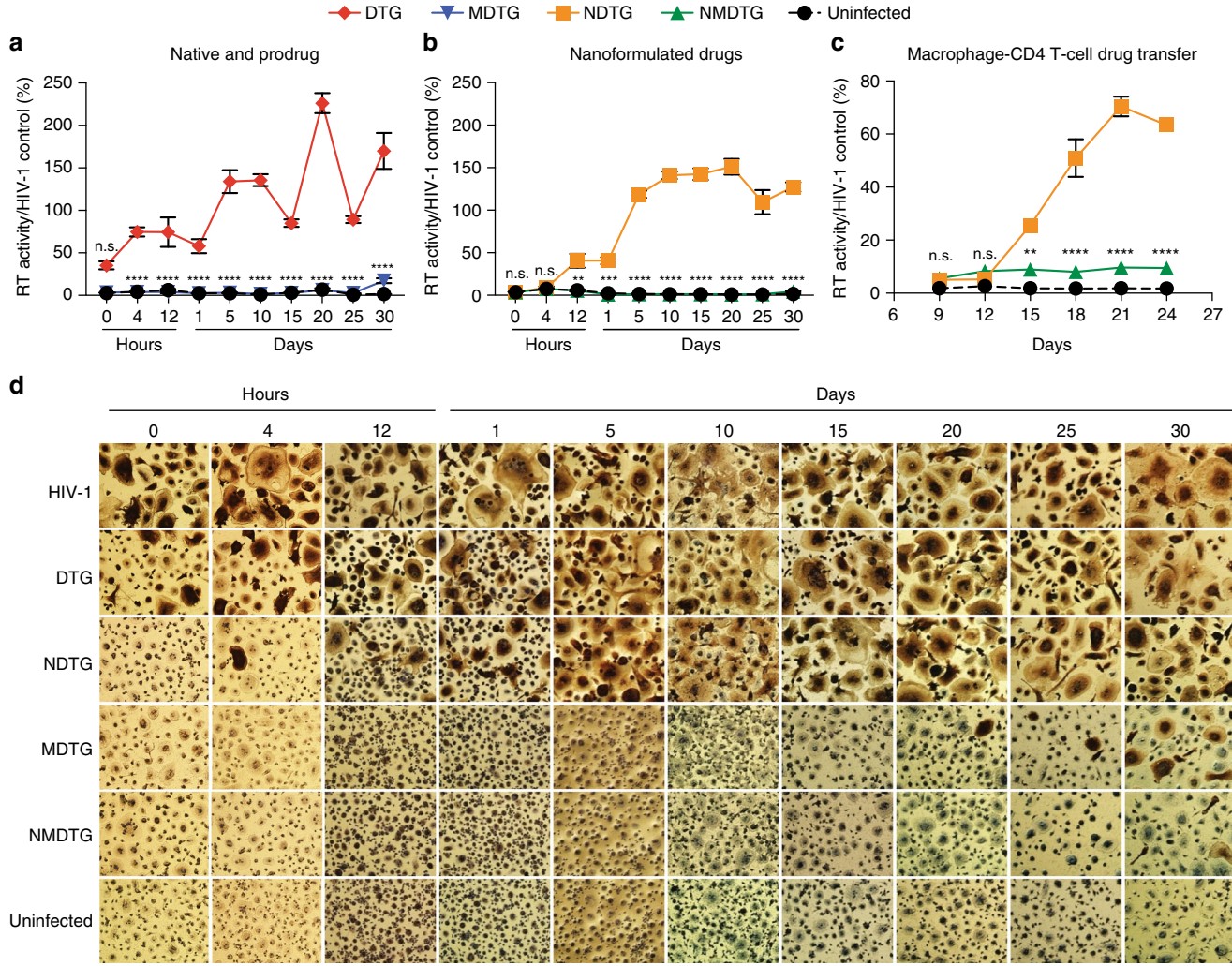

**Fig. 4** Antiretroviral efficacy. **a**, **b** HIV-1 RT activity of (**a**) DTG and MDTG, and (**b**) NDTG and NMDTG-treated MDM. **P** = 0.0039, ***P = 0.0007, ****P < 0.0001 MDTG vs. DTG and NMDTG vs. NDTG. **c** Prevention of spreading viral infection was assessed in human peripheral blood lymphocytes (PBLs) following addition of media conditioned from drug-treated MDM. NMDTG conditioned media significantly reduced HIV-1 RT activity in PBLs compared to NDTG conditioned media beginning at day 15 and maintained protection up to day 24. **P = 0.0018, ****P < 0.0001 NMDTG vs. NDTG. **d** Representative HIV-1p24 staining (brown) of virus-infected MDM-treated with native or nanoformulated drugs are shown. For all, uninfected cells without treatment served as negative controls. HIV-1-infected cells without treatment served as positive controls. Results were normalized to positive control cells. All results are shown as the mean ± SEM of three biological replicates. Results from **a**, **b**, **c** were analyzed by two-way ANOVA with Bonferroni's multiple comparison tests

MDTG was thoroughly characterized by proton nuclear magnetic resonance ($^1$H NMR), carbon nuclear magnetic resonance ($^{13}$C NMR), and Fourier-transform infrared (FTIR) spectroscopy, positive electrospray ionization mass spectroscopy (ESI-MS), and powder X-ray diffraction (XRD). $^1$H NMR spectral analysis demonstrated the loss of the phenol proton peak at 12.5 p.p.m. in the DTG spectrum (Supplementary Fig. 1a). This was accompanied by chemical shifts at 0.9 p.p.m. and 1.26 p.p.m. for MDTG, representing the $1^0$ (C**H$_3$**R) and $2^0$ (RC**H$_2$**R) protons of the aliphatic fatty-acid chain. Chemical shifts at 2.73 p.p.m. and 1.8 p.p.m. correspond to protons at the $C_\alpha$ and $C_\beta$ positions of the aliphatic ester (COOR) alkyl chain (Supplementary Fig. 1b). $^{13}$C NMR spectral analysis of MDTG shows all 34 carbon atoms present in the modified drug (Supplementary Fig. 1c). ESI-MS analysis shows the exact mass of MDTG to be 629.33 (100%), with daughter ion peaks at 630.33 (36.8%) and 631.33 (3.9%) (Supplementary Fig. 1d). Additionally, strong absorption bands at 2915 cm$^{-1}$ and 2850 cm$^{-1}$ in the FTIR spectrum of MDTG

correspond to asymmetric and symmetric methyl C–H stretches of the myristoyl alkyl group. Absorption bands at 1795 cm$^{-1}$ and 1750 cm$^{-1}$ in the spectra of myristoyl chloride and MDTG correspond to carbonyl C=O stretching of the acyl chloride and ester functional groups, respectively (Fig. 1b). XRD confirmed the crystalline forms of both DTG and MDTG, displaying unique diffractin patterns indicating different atomic arrangements within the crystal lattice of each compound (Fig. 1c). Aqueous solubility and IC$_{50}$ were compared for both DTG and MDTG. MDTG exhibited an 8.7-fold decrease in aqueous solubility compared to DTG, confirming the expected enhanced hydrophobicity from the fatty acid conjugate (Fig. 1d). Kinetics of MDTG cleavage were assessed in whole blood. After 60 min, 79.9% of MDTG was cleaved while 88.7% was hydrolyzed in blood diluted ten-fold. By comparison, 15.2% of MDTG was hydrolyzed in blood spiked with an esterase inhibitor and 21.9% was cleaved after acetonitrile (ACN) addition. This changed little over a 300 min observation period (Supplementary Fig. 2e). HIV-

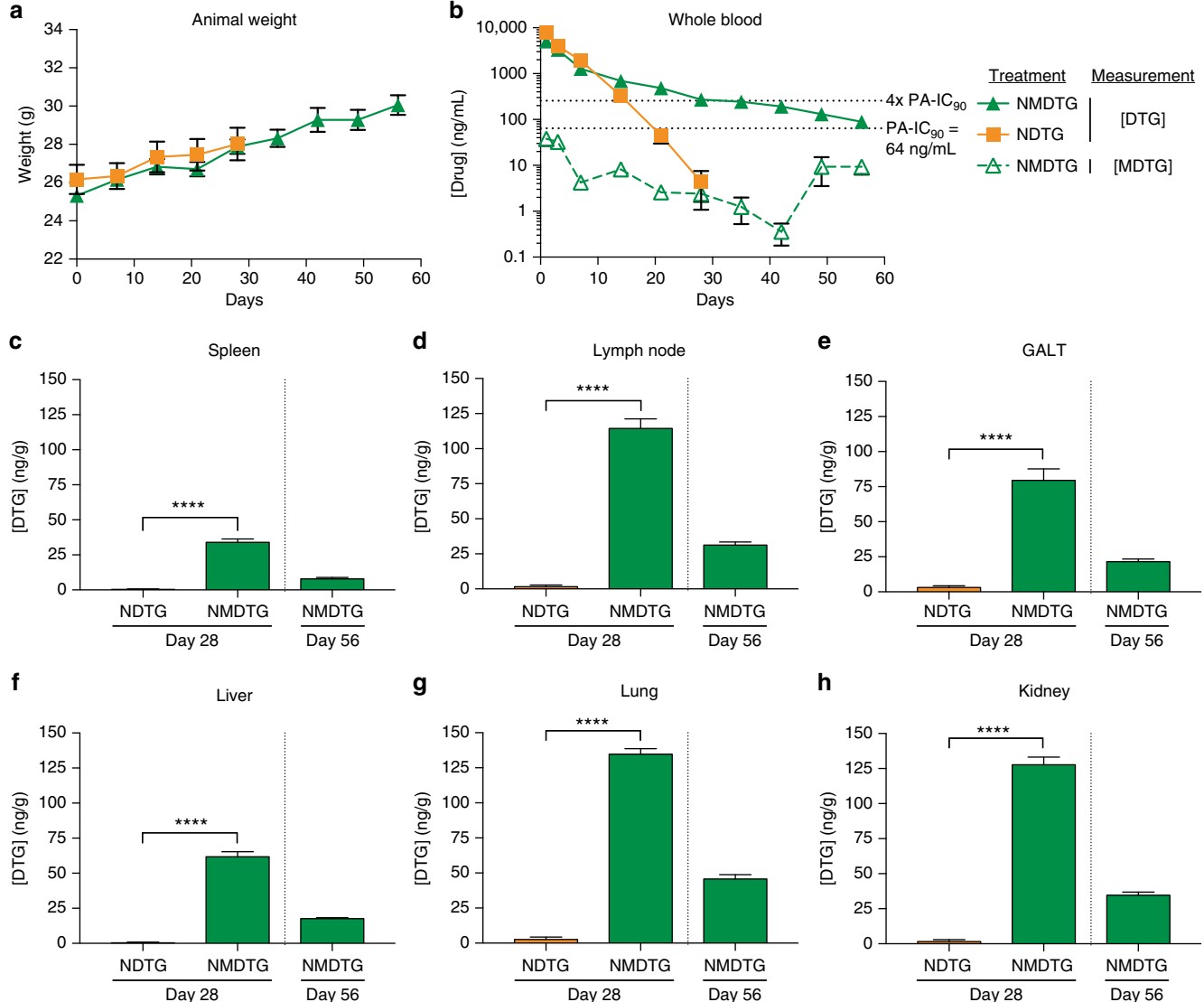

**Fig. 5** Pharmacokinetics. Balb/cJ mice were administered a single IM dose of NDTG or NMDTG (45 mg/kg DTG-eq.) to determine pharmacokinetic (PK) profiles. **a** Animal weights were monitored for the length of the study to assess animal health. **b** Blood DTG and MDTG concentrations were analyzed by UPLC-MS/MS. Solid lines indicate [DTG], while the dashed line indicates [MDTG] from NMDTG treatment. Dotted lines indicate the PA-IC$_{90}$ (64 ng/mL) and four-times the PA-IC$_{90}$ (256 ng/mL). **c-h** Tissue DTG concentrations were analyzed by UPLC-MS/MS. (**c**) Spleen, (**d**) lymph node, (**e**) GALT, (**f**) liver, (**g**) lung, and (**h**) kidney DTG levels are shown at days 28 and 56. ****$P < 0.0001$. Results are shown as the mean ± SEM of at least six biological replicates. Results from **c-h** were analyzed by two-tailed Student's $t$ test (for all, $n = 6$ NDTG, 6 NMDTG, degrees of freedom = 10; **c** $t = 14.4$; **d** $t = 16.4$; **e** $t = 9.3$; **f** $t = 17.3$; **g** $t = 30.1$; **h** $t = 22.7$)

1 reverse transcriptase (RT) activity determined that the IC$_{50}$ of MDTG was not statistically different from DTG (62.5 nM and 56.7 nM, respectively; $P = 0.8397$) (Fig. 1e), confirming that antiretroviral activity of DTG was not altered following modification.

**Nanoformulation stability and release kinetics.** Poloxamer 407 (P407) nanoformulations were prepared using direct synthesis by high-pressure homogenization. Encapsulation efficiencies of NMDTG and NDTG were 82.2 ± 4.4% and 66.2 ± 1.0%, respectively (data not shown). Formulation stability was determined over 134 days at both 4 °C and 25 °C (Fig. 2a–c). Particle size, polydispersity index (PDI), and zeta potential for all formulations were determined by dynamic light scattering (DLS). The size (Fig. 2a), PDI (Fig. 2b), and zeta potential (Fig. 2c) of NMDTG

were 234 ± 2 nm, 0.23 ± 0.02, and −21.6 ± 0.3 mV at day 0, respectively, and remained stable at 4 °C over 134 days (363 ± 4 nm, 0.23 ± 0.01, and −12.1 ± 0.2 mV at day 134). However, NMDTG was less stable at 25 °C, showing differences of 70 nm and 10.0 mV in size and zeta potential at day 134 from day 0, respectively, with no change in PDI. NDTG particles (368 ± 6 nm at day 0) were larger than NMDTG particles, but remained stable over the entire 134 days at both 4 °C and 25 °C (398.1–434.2 nm, 0.22–0.25, and 20.6–25.7 mV at day 134). Particle release kinetics were assessed for the original, neat, undiluted formulations (Fig. 2d) to test for stability of manufactured LASER ART, as well as after 10-fold dilution (Fig. 2e) to test the stability of dosing solutions. For the neat, as well as diluted NDTG, there was a burst release of approximately 80% at time zero, and the entire DTG content was released by day 70 from the neat formulation. Only an additional 5% was released from the diluted NDTG by day 7.

In contrast, for NMDTG, the burst release was only 3.5% and 23% of NMDTG content at day zero for the neat and diluted NMDTG, respectively. No further release of MDTG from the neat formulation was observed over 70 days; whereas, only an additional 5% was released from the diluted formulation over 7 days. In all incubations, mass balance analyses showed that 100% of the drug was recovered.

**Nanoparticle characterization**. Scanning electron microscopy (SEM) was used to assess particle morphology (Fig. 3a, b). NMDTG particles (Fig. 3b; magnification = ×30,000) showed uniform, dominant rod-shaped morphologies, while NDTG particles (Fig. 3a; magnification = ×20,000) were of heterogeneous size and shape and consisted of both cuboidal-shaped and rod-shaped morphologies. The former are known to be more amenable for MDM uptake[16,17]. NMDTG was taken up avidly by MDM and intracellular concentrations increased over a 24-h test period (Fig. 3c). At 24 h, the intracellular drug concentration was 74.3 µg/10$^6$ cells for NMDTG, 185-fold higher than NDTG (0.40 µg/10$^6$ cells) after exposure to an equimolar concentration of DTG. Native MDTG also displayed significantly higher uptake in MDM than either native DTG or NDTG ($P < 0.0001$); however, cellular drug levels reached a maximum at 4 h (16.7 µg/10$^6$ cells). Neither native DTG nor NDTG achieved more than 1.0 µg/10$^6$ cells over 24 h. NMDTG was also retained within MDM for up to 30 days (31 ng/10$^6$ cells) (Fig. 3d). Native DTG and NDTG were at undetectable levels at 24 h (<0.1 µg/10$^6$ cells). DTG, MDTG, and NDTG all showed rapid release from MDM into the surrounding media (Fig. 3e). However, NMDTG displayed a sustained, slow-release profile that reached a maximum at post drug treatment day 5 and continued to day 14. No MDTG was detected with either MDTG or NMDTG treatments, indicating MDTG rapidly hydrolyzes to DTG. Together, these data indicate that NMDTG possesses long-acting and slow-release potential. Even with such high intracellular drug levels, NMDTG showed no toxicity to MDM as determined by 3-(4,5-dimethylthiazol-2-yl)-2,5-diphenyltetrazolium bromide (MTT) assays after 6 and 24 h of drug treatment (Supplementary Fig. 2a and Fig. 3f, respectively). Only 24-h treatment of NDTG at the highest concentration (500 µM) showed reduced cell viability of 44.4%. NMDTG also showed no toxicity as determined by lactate dehydrogenase (LDH) released into media after 24 h of drug treatment at 100 or 400 µM concentrations in MDM. By contrast, 400 µM DTG or NDTG treatment showed changes in cell vitality (Supplementary Fig. 2c). Functionally, MDM exhibited no adverse reactions after NMDTG treatment. Phagocytic function of these cells remained unchanged after 8 h incubation with 10–500 µM of NDTG or NMDTG (Supplementary Fig. 2b). Also, no deleterious reactive oxygen species were detected after 2 h exposure to parent drugs or their nanoformulations (Supplementary Fig. 2d). Transmission electron microscopy (TEM) was used to visualize particles within MDM (Fig. 3g–i). NMDTG can be observed in intracellular compartments within the MDM (Fig. 3i). After 8-h of NMDTG treatment, about 50% of the MDM cytoplasm was comprised of vesicles containing nanoparticles. This was not seen with NDTG as only a few cells showed any intracellular accumulation of nanoparticles (Fig. 3h).

**Antiretroviral efficacy**. To assess antiretroviral activity, HIV-1 RT activity and HIV-1p24 antigen expression were evaluated in infected MDM. Cells were challenged with HIV-1$_{ADA}$ for up to 30 days after a single 8-h treatment with 100 µM drug. Native DTG and NDTG efficacy was observed for up to 4 h after drug treatment (Fig. 4a, b, d). Full inhibition was only measured immediately after NDTG treatment (0 h), with maximal

inhibition of only 65% following native DTG treatment. Native MDTG and NMDTG exhibited improved antiretroviral efficacies compared to their parent drug counterparts. Significantly lower RT activity was detected in media from NMDTG-treated cells compared to NDTG treatment beginning 12 h ($P = 0.0050$), and up to 30 days ($P < 0.0001$), after drug loading, with viral breakthrough beginning at day 30 (96% viral inhibition) (Fig. 4b, d). Native MDTG mirrored these results, with viral breakthrough beginning at day 20 and increasing to day 30 (83% inhibition at day 30) (Fig. 4a, d). HIV-1p24 antigen expression (brown stain) verified all RT results (Fig. 4d).

**Effect of macrophage-released DTG on spreading CD4+T cell infection**. Prevention of spreading HIV-1 infection was assessed in PHA/IL-2-treated PBL (lymphoblasts) after addition of conditioned media from drug-treated MDM. Conditioned media, which contained drug released from MDM during a 24-h period after drug treatment, was used to treat lymphoblasts during spreading HIV-1$_{MN}$ infection. NMDTG conditioned media significantly reduced HIV-1 RT activity in lymphoblasts compared to NDTG conditioned media beginning at day 15 ($P = 0.0018$) (9% vs. 26% RT activity, respectively), and maintained protection up to day 24 ($P < 0.0001$) (10% vs. 63% RT activity, respectively) (Fig. 4c).

**Pharmacokinetics**. Male Balb/cJ mice were administered a single 45 mg/kg DTG-equivalent (equimolar DTG) dose of NDTG or NMDTG intramuscularly (IM) into the caudal thigh muscle to determine pharmacokinetics (PK) over 8 weeks. Whole blood and tissue samples were analyzed by ultra performance liquid chromatography-tandem mass spectroscopy (UPLC-MS/MS) to determine parent and prodrug levels. Neither NDTG nor NMDTG treatments had any adverse effect on animal weight (Fig. 5a). NMDTG displayed a greatly reduced DTG decay curve compared to NDTG, with higher blood drug levels beginning at day 14 (689.8 ng/mL) and extending to day 56 (88.8 ng/mL) (Fig. 5b). At day 28, the blood drug levels were 272.6 ng/mL for NMDTG while NDTG was at or below the limit of quantitation (<4 ng/mL). DTG apparent half-life was increased from 61.9 h for NDTG to 330.4 h for NMDTG (Supplementary Table 1). Similarly, DTG mean residence time (MRT) was more than 3-fold longer with NMDTG than NDTG (104.2 h vs. 348.8 h, respectively). The longer apparent half-life for NMDTG was the result of an approximately 5-fold increase in volume of distribution (V$_\beta$/F) (NDTG–4.55, NMDTG–22.10 L/kg); whereas, clearance (CL/F) was essentially the same compared to NDTG (NDTG–0.051, NMDTG–0.046 L/h/kg). Average blood DTG levels for NMDTG remained above the PA-IC$_{90}$ (64 ng/mL) for the entire 8-week test period, and above four times the PA-IC$_{90}$ for 28 days. DTG concentrations for NDTG-treated mice were above the PA-IC$_{90}$ and four times the PA-IC$_{90}$ for 14 days (328.3 ng/mL). The prodrug was detectable in blood over the first 3 days. At day 28, NMDTG-treated mice had significantly higher DTG levels than NDTG-treated mice in spleen, lymph node, gut-associated lymphoid tissue (GALT), liver, lung, and kidney tissues ($P < 0.0001$). Drug levels in tissues were between 25-fold and 123-fold higher with NMDTG treatment than NDTG at day 28 (GALT and liver, respectively). NMDTG-treated animals maintained detectable drug levels at day 56 (8.0, 31.2, 21.5, 17.6, 45.8, and 34.7 ng/g for spleen, lymph node, GALT, liver, lung and kidney, respectively), while no drug was detected in tissues from NDTG-treated animals (Fig. 5c–h).

**In vivo measures of HIV-1 restriction and protection**. For initial screening of viral restriction by NDTG and NMDTG,

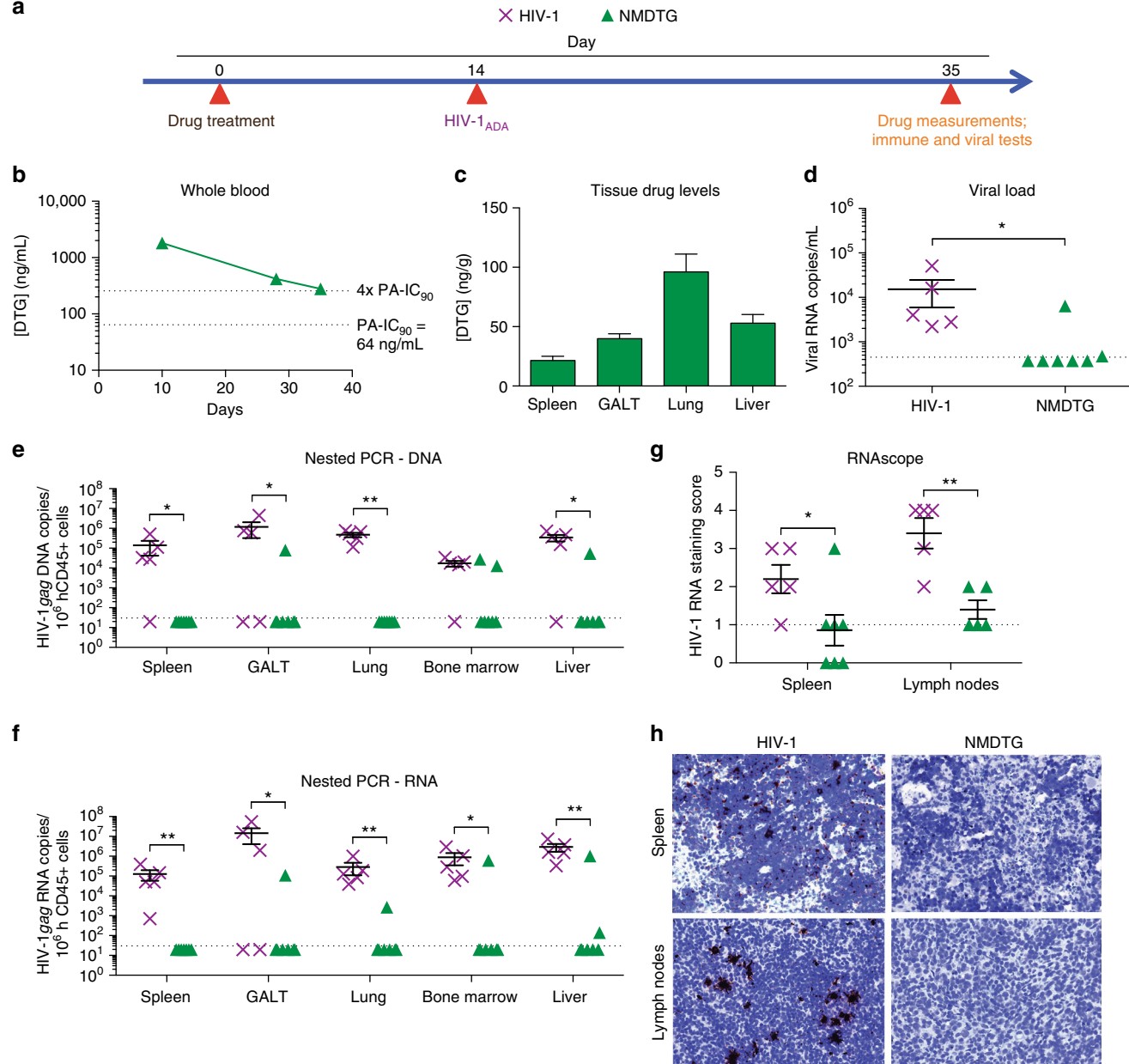

**Fig. 6** Protection against HIV-1 challenge in CD34+ humanized mice. CD34+ hematopoietic stem cell (HSC)- reconstituted NSG mice were treated with NMDTG according to the scheme illustrated in **a**. HIV-1-infected mice without treatment served as positive controls. **b** Blood DTG concentrations were analyzed by UPLC-MS/MS. Dotted lines indicate the PA-IC$_{90}$ (64 ng/mL) and four-times the PA-IC$_{90}$ (256 ng/mL). **c** DTG concentrations were also analyzed in spleen, GALT, lung, and liver samples. **d** Plasma viral load was measured three-weeks after HIV-1 challenge. **e** DNA and **f** RNA semi-nested real-time PCR was performed on spleen, GALT, lung, bone marrow, and liver. **g** HIV-1 RNAscope was performed on spleen and lymph node sections and scored according to amount of positive staining [0 = no staining or <1 dot/10 cells, 1 = 1–3 dots/cell, 2 = 4–9 dots/cell with no, or very few, dot clusters, 3 = 10–15 dots/cell and <10% dots are in clusters, 4 = >15 dots/cell and >10% dots are in clusters]. Anything scoring less than or equal to one was considered as background. **h** Representative HIV-1 RNAscope staining (brown) of spleen (top) and lymph node (bottom) sections are shown. *$P$ < 0.05, **$P$ < 0. 01. Results are shown as the mean ± SEM of five positive control (5 female) and 7 NMDTG-treated (5 female, 2 male) animals. Results were analyzed by two-tailed Student's $t$ test ($n$ = 5 HIV-1 control, 7 MDTG; degrees of freedom = 10)

NOD/scid-IL-2R$\gamma$c$^{null}$ (NSG) mice were reconstituted by intraperitoneal (IP) injection of human peripheral blood lymphocytes (hu-PBLs) 4 or 19 days post drug treatment generating hu-PBL-NSG, mice. Mice were administered a single IM dose of NDTG or NMDTG at a concentration of 45 mg/kg DTG-eq. on day 0 (Supplementary Fig. 3a) then challenged 10 days post hu-PBL reconstitution with 2 × 10$^4$ TCID$_{50}$ of HIV-1$_{ADA}$ administered by IP injection. Animals were euthanized 10 days post HIV-1 challenge, and blood, plasma, and tissues were collected for drug levels, human T cell counts, viral load, and HIV-1p24 expression. Neither NDTG nor NMDTG treatment adversely affected animal weights (Supplementary Fig. 4a, e). Enhanced levels of viral restriction were observed by NMDTG over NDTG treatments when mice were challenged two or four weeks after a single IM drug dose. Levels of viral reductions were 3.0-log$_{10}$ and 1.2-log$_{10}$ in plasma viral load compared to HIV-1 infected controls and

1.8-$\log_{10}$ and 1.1-$\log_{10}$ decreases in plasma viral load for NMDTG-treated mice compared to NDTG-treated mice, when challenged two weeks or four weeks post-treatment, respectively (Supplementary Fig. 3b, d). HIV-1p24 expression was also determined in paraffin-embedded spleen sections. When challenged two weeks post-treatment, NMDTG provided restriction of viral infection in spleen (Supplementary Fig. 3c, f). NMDTG treatment also reduced viral p24 expression in spleen when challenged four weeks post-treatment (4.4% of HLA-DP/DQ/DR + cells compared to 15.6% and 14.6% for HIV-1 infected controls ($P = 0.0006$) and to NDTG ($P < 0.0025$; Supplementary Fig. 3e, g). With NMDTG treatment, average blood DTG levels in the hu-PBL-NSG mice remained above the PA-$IC_{90}$ (64 ng/mL) for the entire 52-day test period (125.5 ng/mL) and above four times the PA-$IC_{90}$ for 28 days (275.3 ng/mL; Supplementary Fig. 3 h). NDTG treatment remained above the PA-$IC_{90}$ for 28 days (78.3 ng/mL) and above the four times the PA-$IC_{90}$ for up to 21 days (220.4 ng/mL). Tissue DTG concentrations in spleen, GALT, liver, lung, and kidney are shown at days 24, 39, and 52 (Supplementary Fig. 3i-m). Drug levels in tissues were on average 2.1-fold higher after NMDTG treatment than NDTG on day 24. Little to no drug was detected in tissues from DTG-treated animals beginning at day 39, with only kidney having measurable drug levels (4.1 ng/g). NMDTG-treated animals maintained detectable drug levels through day 52.

As graft-vs-host disease induces CD4+ T cell activation in hu-PBL mice, a second murine model was employed to assess in vivo efficacy of NMDTG against viral challenge. This was used to both extend and validate the initial tests. Here, the ability of NMDTG to protect against a viral challenge of $2 \times 10^4$ $TCID_{50}$ HIV-1$_{ADA}$ was determined in NSG mice reconstituted at birth with CD34+ human hematopoietic stem cells (HSC). HSC were obtained from human cord blood. Importantly, the reconstituted mice show quiescent lymphocyte profiles and as such reflect what would occur in a HIV-1 challenged human host. Thus, these mouse validation tests would more closely mimic what would occur during pre-exposure prophylaxis (PrEP). NMDTG was the sole tested formulation for the longer-term protection studies since PK tests for NDTG showed drug levels below the PA-$IC_{90}$ by 21 days and no detectable drug by day 28 (Fig. 5b–h). NDTG treatment also recorded limited long-term viral restriction (Supplementary Fig. 3). Thus, for these mice, a single IM dose of NMDTG was administered at a concentration of 45 mg/kg DTG-eq. at 22 weeks of age (Fig. 6a). Mice were challenged two weeks following drug treatment with $2 \times 10^4$ $TCID_{50}$ of HIV-1$_{ADA}$ by IP injection. Mice were bled three weeks post HIV-1 challenge; and blood and plasma collected to assess drug levels and viral loads. Neither NMDTG treatment nor viral challenge had an adverse effect on animal weight throughout the experimental period (data not shown). Average blood DTG levels remained above four times the PA-$IC_{90}$ (279.1 ng/mL; 4.4 times the PA-$IC_{90}$ at day 35) for the entire length of the study (Fig. 6b). Tissue drug levels in spleen, GALT, lung, and liver were concordant with previous experiments at day 35 (Fig. 6c). NMDTG-treated mice showed protection when challenged with virus two weeks after a single IM dose; with detectable plasma viral load in a single NMDTG-treated animal and one at the limit of detection (Fig. 6d). Semi-nested real-time PCR confirmed protection against viral challenge with cell-associated HIV-1*gag* DNA and RNA in NMDTG-treated spleen, GALT, lung, bone marrow, and liver below the limit of detection (Fig. 6e, f). HIV-1 RNAscope was performed on spleen and lymph node sections, and HIV-1 RNA staining was scored according to the manufacturer's pre-determined criteria in a blinded manner (Fig. 6g, h). Due to high technique sensitivity (detection limit of 1 viral RNA copy/cell using 78 probes spanning 7437 base pairs) a score of <1 was considered as background. HIV-1 RNA staining score for both HIV-1-infected and NMDTG-treated spleen (2.2 vs 0.86, respectively; $P = 0.0418$) and lymph node tissue sections (3.4 vs. 1.4, respectively; $P = 0.0027$) demonstrated NMDTG protection.

## Discussion

A key part of any effective antiretroviral regimen rests in ensuring that patients take their prescribed medicines[3,18]. HIV/AIDS treatment regimens are currently defined by daily or twice-daily dosing intervals[19]. Adherence underlies clinical responsiveness and any consequent emergence of viral resistance. It also affects the accompanying stability of CD4+ T cell numbers and function[4]. Thus, any ART regimens that allow infrequent dosing with an ability to maintain consistent drug levels in plasma and tissues sufficiently above the $IC_{90}$ would improve clinical outcomes and hold viral replication in check[5,6]. Particle size and physicochemical properties were used to optimize release of drug molecules from delivery systems. Nanoparticle based systems are readily taken up by cells and disseminated into tissues to form drug depots at these sites for subsequent slow release[20]. Other controlled release injection site drug depot formulations include microparticle carrier systems[21,22]. However, microparticles tend to aggregate limiting their usage due to lack of particle homogeneity, leading to injection site reactions[23].

In the current study, we chemically modified DTG enabling the creation of poloxamer-encased hydrophobic and lipophilic drug nanocrystals. Prodrugs can offer therapeutic benefits over native compounds by providing reduced drug metabolism and toxicity. They may also increase lipophilicity and thus improve cell membrane and tissue permeability of drug[24]. Proof of concept for such advances are highlighted by antipsychotic drugs. Indeed, these have become widely used as long-acting hydrophobic ester prodrugs[25]. Likewise, the creation of NMDTG improved drug to polymer interactions to form stable nanocrystals and boosted delivery of the nanoformulated drug into MDM autophagosomes[26]. The MDTG nanocrystals undergo slow intracellular dissociation within endosomes and prodrug cleavage to protect the cell against viral challenge for up to or beyond 25 days. PK and PD evaluations showed significant improvements in drug apparent half-life, biodistribution, and antiretroviral activities over the native drug. Altogether, the creation of an ester prodrug of DTG using myristic acid enabled intrinsic drug crystal formation and bioconversion. These reactions occurred in the presence of biological fluids containing esterases yielding pharmacologically active medicines[15].

No mouse model exactly reflects PrEP, thus we used both viral restriction and protection murine models in this study. It is noteworthy that the evaluation of viral restriction as performed in hu-PBL mice failed to provide protection against viral challenge by less than one month, as shown by detectable plasma viral load. This likely reflects enhancement in viral susceptibility in the animals based on the large numbers/percentages of activated lymphocytes due to xenoreactivity leading to graft-vs-host disease, timing of human cell reconstitution, larger viral challenge, route of viral challenge, and monotherapy approach[27−30]. CD34+ HSC reconstituted NSG mice better reflect human biology as there is no graft-vs-host disease and the reconstitution contains monocyte-macrophages, as well as CD4+ and CD8+ lymphocytes[31]. To these ends, this model was used for confirmatory studies. Indeed, cells were quiescent prior to viral challenge. Most importantly, NMDTG-treated HSC-NSG mice demonstrated plasma viral load, as well as tissue viral copies, below the limit of detection in five of seven animals for two weeks against HIV-1 challenge following a single nanoparticle injection. Due to

biological limitations in the animal models used, protection in all animals was not achieved. These results reflect what was observed in humans who are at risk of viral infection despite optimal PrEP treatment[32]. Due to experimental limitations, we also cannot exclude the possibility that high concentrations of plasma drug could suppress HIV-1 breakthrough infection. Based on the intrinsic properties of drug stability and with retention in MDMs in tissues, such chemical prodrug modifications led to the formation of a second drug reservoir beyond the injection site. Such improvements in antiretroviral drug structure and packaging can not only improve drug adherence, but also could reduce systemic toxicities. NMDTG PK studies in non-human primates further validated these results[33]. Three male rhesus macaques were administered a single IM dose of NMDTG at a concentration of 25.5 mg/kg DTG-eq. Plasma DTG concentrations remained above the PA-IC$_{90}$ for 35 days and increased DTG apparent half-life to 467 h. Notably, this came with no alterations in neutrophil, lymphocyte, or monocyte counts, animal weights, or liver and kidney metabolic profiles. This study further demonstrates that a single IM injection of long-acting NMDTG can provide plasma levels above the PA-IC$_{90}$ for one month and can greatly extend drug apparent half-life, with no signs of toxicity.

Long-acting drug formulations have been extensively investigated, including their prolonged apparent half-life, high protein-binding, lipid or surfactant drug nanocrystal encasement, rapid drug entry into monocyte-macrophages, and slow drug release[34]. These long-acting properties are mostly due to enhanced particle/ drug stability, rapid cell and barrier penetration, and/or slowed intracellular drug hydrolysis[24]. Cumulatively, these properties enhance therapeutic responses, allow for cell-mediated drug delivery, and enable improved delivery of drugs to areas of poor drug penetrance and viral reservoirs[35]. One of the most important of these is uptake and sequestration into MDM. Highly mobile MDM have large storage capacities and can act as Trojan Horses for drug delivery to circulating and tissue CD4+ T cells and viral reservoirs[36]. This is especially operative when describing lymphoid organs, where macrophages and T cells are in intimate contact, permitting passage of drug to such major reservoirs of HIV infection[37–39]. Here, we demonstrate that drug released from NMDTG-treated MDM can act upon T cells and significantly inhibit and prevent spreading of viral infection within T cell cultures. Macrophages can protect drugs from metabolism— prolonging apparent half-life—and enhance efficacy, PK, and biodistribution of the drug delivery system[40–42]. Once inside the macrophage, drug particles can be stored in late- and recycling-endosomes, as well as autophagosomes and as such, create a secondary drug depot within the tissue macrophage independent of the muscle site of injection[26,43,44]. Indeed—and in addition to the macrophage's phagocytic, clearance, antigen presentation, and secretory functions—the cells also serve as viral sanctuaries, vehicles for viral transport, and as reservoirs for HIV-1 replication[45,46]. Thus, delivery of drug to macrophages can serve multiple critical purposes.

In clinical settings, DTG rarely elicits viral resistance mutations in infected patients when used as first-line therapy, with suboptimal adherence driving these mutations[47]. In vitro, the most common mutation against DTG is the R263K that severely reduces viral replication fitness, reducing the impact and rendering it insignificant for HIV/AIDS treatments[48]. Prevention of renewed viral infection emerging from tissue reservoirs experiencing suboptimal drug levels may allow infected cells to die off through normal cell turnover without spreading virus to other cells that may act as viral reservoirs[49]. Thus, the changes in antiretroviral drug treatment made here could limit viral replication in its native reservoirs, allowing antiretroviral drugs alone to keep the virus in check and attenuate new infections. The

offering of sustained high plasma and tissue drug levels in time periods measured in months, without notable drug peaks and troughs, could also enable more efficient excision of integrated proviral DNA through CRISPR/Cas9 technologies[50]. Currently under evaluation for clinical use are transdermal patches and ARV implants. Such devices, including dapivirine vaginal rings, subdermal silicone tenofovir alafenamide and polymeric EFdA, are being developed for PrEP[51–53]. Refillable channel devices capable of loading multiple drugs have also been described[54]. These devices could be loaded with NMDTG and/or other LASER ART nanocrystals to further extend the apparent half-lives of the parent medicines with a single implant. LASER ART-loaded microneedle patches similar to those used to deliver injectable contraceptives could also be used to alleviate pain during administration[55].

The concept of long-acting injectable drugs is well received by patients with HIV, with 73% indicating that they would definitely or probably try them[56]. This number goes up to 84% when patients were asked about monthly dosing as opposed to weekly or biweekly dosing. Conceivably, bimonthly or longer dosing intervals would be even more attractive. Future endeavors to further extend drug apparent half-lives and better target macrophages, including further drug modifications, linking two drug molecules to a single linker, combining divergent classes of antiretroviral drugs, and nanoparticle surface decorations will shape the future of long-acting antiretroviral therapy. Future works will focus on refinements in drug modification and formulation, and the packaging of multiple drugs into a single formulation for delivery. Strategies such as these will further improve adherence to drug regimens and biodistribution profiles, and maintain high drug levels for even longer periods of time, limiting viral resistance mutations and, overall, improving ART efficacy and patient outcomes[33,57,58].

## Methods

**Reagents**. DTG was a generous gift from ViiV Healthcare (Research Triangle Park, NC). Pyridine, dimethylformamide (DMF), N,N-diisopropylethylamine (DIEA), myristoyl chloride, poloxamer 407 (P407), 4-(2-hydroxyethyl)-1-piper-azineethanesulfonic acid (HEPES) buffer, ciprofloxacin, 3-(4,5-dimethylthiazol-2-yl)-2,5-diphenyltetrazolium bromide (MTT), dimethyl sulfoxide (DMSO), paraf-ormaldehyde (PFA), and 3,3'-diaminobenzidine (DAB) were purchased from Sigma-Aldrich (St. Louis, MO). Diethyl ether, cell culture grade water (endotoxin-free), gentamicin, acetonitrile (ACN), methanol, KH$_2$PO$_4$, bovine serum albumin (BSA), Triton X-100, LC-MS-grade water, and TRIzol reagent were purchased from Fisher Scientific (Hampton, NH). FITC mouse anti-human CD45, Alexa Fluor 700 mouse anti-human CD3, APC mouse anti-human CD4, BV421 mouse anti-human CD8, PE mouse anti-human CD14, and PE-Cy5 mouse anti-human CD19 were purchased from BD Biosciences (San Jose, CA). Monoclonal mouse anti-human HIV-1p24 (clone Kal-1), monoclonal mouse anti-human leukocyte antigen (HLA-DP/DQ/DR; clone CR3/43), and the polymer-based HRP-conjugated anti-mouse EnVision+ secondary were purchased from Dako (Carpinteria, CA). Heat-inactivated pooled human serum was purchased from Innovative Biologics (Herndon, VA). Dulbecco's Modification of Eagle's Medium (DMEM) was purchased from Corning Life Sciences (Tewksbury, MA).

**MDTG synthesis and characterization**. A lipophilic and hydrophobic modified DTG prodrug (called MDTG) was synthesized through myristoylation of the parent drug hydroxyl group. Initially, DTG was dried by co-evaporation from anhydrous pyridine then resuspended in anhydrous DMF and cooled to 0 °C under argon. 2-equivalents DIEA were used to deprotonate the 7-hydroxyl group of DTG, which was then immediately reacted with 2 equivalents myristoyl chloride for 24 h. The resultant product was purified by silica gel column chromatography using an initial mobile phase of 4:1 ethyl acetate: hexanes for 12 fractions, then 9:1 ethyl acetate: hexanes for the remainder. Once purified, fractions containing the UV active prodrug were dried and precipitated from diethyl ether. The precipitate was collected by centrifugation at 3500 r.p.m. for 15 min and dried under vacuum, while the supernatant was discarded. MDTG was synthesized at a final drug yield of 82.8%. Proton nuclear magnetic resonance ($^1$H NMR), carbon nuclear magnetic resonance ($^{13}$C NMR), and Fourier-transform infrared (FTIR) spectroscopy, positive electrospray ionization mass spectroscopy (ESI-MS), and powder X-ray diffraction (XRD) were used to characterize the structure of MDTG. NMR was performed on a Bruker Avance-III HD (Billerica, MA) operating at 500 MHz, a

magnetic field strength of 11.7 T. MDTG $^1H$ NMR spectrum specifics: (500 MHz, CDCl$_3$) $\delta$ 10.20 (s, 1 H), 8.45 (s, 1 H), 7.35 (dd, $J$ = 15.0, 8.2 Hz, 1 H), 6.83 (app. dd, $J$ = 19.1, 9.3 Hz, 1 H), 5.26 (br. s, 1 H), 4.85–5.01 (m, 1 H), 4.62 (br. s, 1 H), 4.30 (app. d, $J$ = 12 Hz, 2 H), 4.17 (dd, $J$ = 13.3, 5.9 Hz 1 H), 4.0 (app. d, $J$ = 6.3 Hz, 1 H), 2.73 (t, $J$ = 7.6 Hz, 2 H), 2.17 (td, $J$ = 14.5, 7.2 Hz, 1 H), 1.80 (app. t, $J$ = 7.5 Hz, 2 H), 1.51–1.61 (m, 2 H), 1.40–1.49 (m, 2 H), 1.36 (d, $J$ = 7.0 Hz, 3 H), 1.26 (br. s, 20 H), 0.90 (t, $J$ = 6.6 Hz, 3 H). $^{13}C$ NMR spectrum specifics: (125 MHz, CDCl$_3$) $\delta$ 171.9, 171.1, 163.3, 163.1, 161.6, 161.1, 159.6, 130.6, 130.5, 129.2, 121.3, 121.2, 119.5, 111.2, 111.1, 103.9, 103.7, 103.5, 76.1, 53.1, 36.5, 33.8, 31.9, 29.7, 29.6, 29.5, 29.4, 29.3, 29.2, 29.0, 24.4, 22.6, 15.8, 14.1. ESI-MS specifics ($m/z$): calculated for C$_{34}$H$_{45}$F$_2$N$_3$O$_6$, 629.33 (100%), 630.33 (36.8%), 631.33 (3.9%); found, 630.30. FTIR was performed on a PerkinElmer universal attenuated total reflectance (UATR) Spectrum Two (Waltham, MA). XRD was performed in the 2$\theta$ range of 2–70° using a PANalytical Empyrean diffractometer (Westborough, MA) with Cu-K$\alpha$ radiation (1.5418 Å) at 40 kV and 45 mA and a solid state PIXcel3D detector (Westborough, MA) at a rate of 0.033 °/s with a diffracted beam monochromator. The aqueous solubility of DTG and MDTG was evaluated by adding excess drug to water at room temperature then mixing it overnight. Samples were spun at 14,000 r.p.m. for 10 min to pellet any insoluble drug. Solubilized drug in the supernatant was extracted in methanol and measured using a Waters ACQUITY ultra performance liquid chromatography (UPLC) H-Class System with TUV detector and Empower 3 software (Milford, MA). For DTG and MDTG quantitation, drug extracts were separated on a Phenomenex Kinetex 5 $\mu$m C18 column (150 × 4.6 mm) (Torrance, CA) using either 65% 50 mM KH$_2$PO$_4$, pH 3.2/35% ACN (DTG) or 90% ACN/10% water (MDTG) with a flow rate of 1.0 mL/min and detected at 254 nm and 230 nm, respectively. Drug content was quantitated by comparison of peak area to those of known standards (0.05–50 $\mu$g/mL). Ex vivo cleavage kinetics of MDTG was assessed in mouse whole blood. Ten microliter of 500 ng/mL spiking solution (50 ng/mL final drug concentration) was spiked into 100 $\mu$L blood, blood diluted 10× in PBS, blood that was added to ACN, or blood spiked with an esterase inhibitor cocktail [20 mg/mL sodium fluoride (NaF) and 6 mg/mL ethylenediaminetetraacetic acid (EDTA) with 100 $\mu$M phenylmethylsulfonyl fluoride (PMSF)] and incubated at room temperature. At collection time points 1 mL ACN was added to stop any enzymatic activity. For initial time points, ACN was first added to blood before spiking solution. Samples were then dried and analyzed for MDTG levels by UPLC tandem mass spectrometry (UPLC-MS/MS; see below).

**Nanoparticle synthesis and characterization.** DTG and MDTG nanoparticles (NDTG and NMDTG, respectively) were formulated on an Avestin EmulsiFlex-C3 high-pressure homogenizer (Ottawa, ON, Canada) using P407 to encase the drug crystals. For NDTG, P407 (0.06% w/v) was first dissolved in endotoxin-free water at pH 7.0. Drug (1% w/v) was then added at 100:6 drug–polymer ratio and mixed to form a pre-suspension. For NMDTG, P407 (0.1% w/v) was first dissolved in 10 mM HEPES buffer, pH 7.8. Next, drug (1% w/v) was added at 10:1 drug–polymer ratio and mixed to form a pre-suspension. For both, high-pressure homogenization (~20,000 psi) was then used to generate final homogenous drug nanosuspensions of approximately 250–350 nm. Particle size, polydispersity index (PDI), and zeta potential were determined by dynamic light scattering (DLS) using a Malvern Nano-ZS (Worcestershire, UK)[59]. Particle release kinetics were determined for the original undiluted nanoformulation batches and 10-fold diluted batches with the respective buffers used for manufacture. Diluted and undiluted nanoformulations were incubated over 7 days and 70 days, respectively, at 4 °C. Ten microliter aliquots were collected into 990 $\mu$L of 4% BSA in PBS at various time points. Samples were further diluted to 2 $\mu$g/mL with the same 4% BSA solution, and 10 $\mu$L was withdrawn for total drug concentration analysis. The remaining solution was centrifuged at 10,000×$g$ for 10 min. After centrifugation, 50 $\mu$L aliquots of supernatant were collected in 250 $\mu$L of methanol for released drug concentration analysis. All samples were analyzed by LC-MS/MS for total and released drug concentrations.

**In vitro monocyte-derived macrophage assays.** Human monocytes were obtained by leukapheresis from HIV-1/2 and hepatitis B seronegative donors, and then purified by counter-current centrifugal elutriation[60]. Human monocytes were plated in a 12-well plate at a density of 1.0 × 10$^6$ cells per well using DMEM supplemented with 10% heat-inactivated pooled human serum, 1% glutamine, 10 $\mu$g/mL ciprofloxacin, and 50 $\mu$g/mL gentamicin. Cells were maintained at 37 °C in a 5% CO$_2$ incubator. After 7 days of differentiation in the presence of 1000 U/mL recombinant human macrophage colony stimulating factor (MCSF), MDM were treated with 100 $\mu$M DTG, MDTG, NDTG, or NMDTG. Native drugs were added in DMSO (0.1% v/v). Uptake of drug was assessed by measurements of intracellular drug concentrations at 2, 4, 8, 12, 16, or 24 h after treatment[59]. For drug retention studies, cells were treated for 8 h then washed with PBS and maintained with half-media changes every other day until collection at days 1, 5, 10, 15, 20, and 30. For both studies, adherent MDM were washed with PBS, then scraped into PBS, and counted at indicated time points using an Invitrogen Countess Automated Cell Counter (Carlsbad, CA). Cells were pelleted by centrifugation at 3000 r.p.m. for 8 min at 4 °C. Cell pellets were briefly sonicated in 200 $\mu$L methanol to extract drug and centrifuged at 14,000 r.p.m. for 10 min at 4 °C to pellet cell debris. DTG and MDTG drug content was determined by UPLC-UV/Vis as described above. Release was assessed by collecting culture media after 4-h of drug treatment on days 1, 3, 5,

7, 10, 12, and 14. After collection, 150 $\mu$L of media was added to 1 mL of HPLC-grade methanol and vortexed. Samples were centrifuged at 14,000 r.p.m. for 10 min at 4 °C. Supernatants were dried using a ThermoScientific Savant Speed Vacuum (Waltham, MA), extracted drug resuspended in 150 $\mu$L HPLC-grade methanol, and drug content was determined by UPLC-UV/Vis. To assess cell viability, MTT assay was performed. Briefly, MDM were seeded on 96-well plates at a density of 80.0 × 10$^5$ cells per well and treated with various concentrations (1–500 $\mu$M) of NDTG or NMDTG for 6 or 24 h. After drug treatment, cells were washed and incubated with 100 $\mu$L/well of MTT solution (5 mg/mL) for 45 min at 37 °C. After incubation MTT was removed, and 200 $\mu$L/well of DMSO was added and mixed thoroughly. Absorbance was measured at 490 nm on a Molecular Devices SpectraMax M3 plate reader with SoftMax Pro 6.2 software (Sunnyvale, CA). ROS species were measured in MDMs using a DCFDA cellular ROS detection assay kit as per the manufacturer's instructions (Abcam, Cambridge, MA). Briefly, MDMs were seeded on black, clear-bottom 96-well plates at a density of 80.0 × 10$^5$ cells per well. Cells were then incubated for 45 min in 1× Buffer (supplied with the kit) containing 25 $\mu$M DCFDA at 37 °C, then washed with 1× Buffer. Cells were treated with DTG, MDTG, NDTG, or NMDTG at 100 and 400 $\mu$M for 2 h, and DCF production was measured by fluorescence spectroscopy with excitation and emission wavelengths of 485 nm and 535 nm, respectively. LDH cytotoxicity assay was performed in MDMs using a LDH assay kit as per the manufacturer's instructions (Abcam, Cambridge, MA). Briefly, MDMs were seeded on white, clear-bottom 96-well plates at a density of 80.0 × 10$^5$ cells per well. Cells were then treated with DTG, MDTG, NDTG, or NMDTG at 100 and 400 $\mu$M concentrations for 24 h. Five microliter of media from each well was then mixed with 95 $\mu$L of the reaction mixture (supplied with the kit), followed by measurement of fluorescence at excitation and emission wavelengths of 535 nm and 587 nm, respectively. Phagocytic activity was assessed in MDMs using the Vybrant$^{TM}$ phagocytosis assay kit as per the manufacturer's instructions (Invitrogen, Carlsbad, CA)[61]. Briefly, MDMs seeded on 96-well plates at a density of 80.0 × 10$^5$ cells per well were treated with NDTG or NMDTG over a range of concentrations (10–500 $\mu$M) for 8-h. Cells were then washed with PBS and incubated for 2 h with fluorescently labeled *E. coli* particles (supplied with the kit) at 37 °C. Unbound particles were removed by aspiration, followed by quenching of extracellular *E. coli* with trypan blue for 1 min, and fluorescence measurement at excitation and emission wavelengths of 480 nm and 520 nm, respectively.

**Scanning and transmission electron microscopy.** Nanoparticle morphology was analyzed by scanning electron microscopy (SEM). Briefly, nanosuspensions were air dried onto a glass coverslip mounted on an SEM sample stub and sputter coated with approximately 50 nm of gold/palladium alloy. Samples were examined using a FEI Quanta 200 scanning electron microscope (Hillsboro, OR) operated at 5.0 kV. Drug loaded MDM were analyzed by transmission electron microscopy (TEM) after treatment for 8 h. Cells were washed, scraped into PBS, pelleted at 3000 r.p.m. for 8 min at room temperature, and fixed in a solution of 2% glutaraldehyde, 2% paraformaldehyde in 0.1 M Sorenson's phosphate buffer (pH 6.2). A drop of the fixed cell suspension was placed on a formvar/silicon monoxide 200 mesh copper grid, allowed to settle for 2 min, and the excess solution wicked off and allowed to dry. A drop of NanoVan vanadium negative stain was placed on the grid for 1 min, then wicked away and allowed to dry. Grids were examined on a FEI Tecnai G2 Spirit TWIN transmission electron microscope (Hillsboro, OR) operated at 80 kV, and images were acquired digitally with an AMT digital imaging system (Woburn, MA).

**Antiretroviral activities.** Antiretroviral efficacy was determined by measurements of HIV reverse transcriptase (RT) activity. For IC$_{50}$ determination, MDM were exposed to various concentrations (0.01–1000 nM) of DTG or MDTG for 1 h followed by challenge with HIV-1$_{ADA}$[60] at a multiplicity of infection (MOI) of 0.1 infectious particles per cell for 4 h. Following viral challenge, cells were washed and incubated with the same concentration of drug used before infection for an additional 10 days in culture. Culture fluids were collected on day 10 for the measurement of RT activity as previously described[16,62,63]. To assess antiretroviral efficacy, MDM were treated with 100 $\mu$M DTG, MDTG, NDTG, or NMDTG as described above for 8 h. After treatment, cells were washed with PBS and cultured with fresh media, with half-media exchanges every other day. At 0, 4, 12 h, and 1, 5, 10, 15, 20, 25, or 30 days after treatment, cells were challenged with HIV-1$_{ADA}$ at an MOI of 0.1 infectious particles per cell for 16 h. After viral infection, the cells were cultured an additional 10 days with half-media exchanges every other day. Culture fluids were collected for measurement of RT activity as previously described[16,62,63]. Cells were fixed with 4% PFA and expression of HIV-1p24 antigen was determined by immunocytochemistry.

**Effect of macrophage-released DTG on T cell infection.** Human MDM were treated with 100 $\mu$M NDTG or NMDTG for 4 h, as described above. Following 4-h treatment, cells were washed and fresh media was applied for 24 h. Conditioned medium, containing drug released from MDM during this 24-h period, was collected and used to assess antiretroviral activity in human peripheral blood lymphocytes (PBLs). Freshly elutriated PBLs were stimulated with 10 $\mu$g/mL mitogen phytohemagluttinin (PHA) and 20 U/mL interleukin-2 (IL-2) for 2-3 days. PBLs were then infected with HIV-1$_{MN}$ at an MOI of 0.1 for 8 h in the presence of

conditioned media from drug-treated MDM. Cells were washed and cultured with fresh media. PBLs were replenished every three days to replace any dead and/or dying cells due to HIV-1 infection. At days 9, 12, 15, 18, 21, and 24 after viral challenge, culture fluids were collected for the measurement of RT activity as previously described[16,62,63].

**Immunocytochemistry**. For immunocytochemistry, cells were washed with PBS and fixed with 4% PFA at room temperature for 15 min, followed by an additional washing with PBS. The cells were treated with a blocking/permeabilizing solution (1% Triton X-100, 10% BSA in PBS) and incubated with mouse monoclonal antibody to HIV-1p24 (1:100) for 16 h at 4 °C. Cells were washed with PBS, and polymer-based HRP-conjugated anti-mouse secondary was added for 30 min at room temperature. Cells were then washed with PBS and developed with DAB. Nuclei were counterstained with Mayer's hematoxylin, and cells were visualized with a ×20 objective on a Nikon Eclipse E800 microscope (Melville, NY) with Nuance EX multispectral imaging system (PerkinElmer, Hopkinton, MA)[16,59].

**Pharmacokinetics**. Male Balb/cJ mice (6–8 weeks of age; Jackson Labs, Bar Harbor, ME) were injected with NDTG or NMDTG (45 mg/kg DTG-eq.) intramuscularly (IM; caudal thigh muscle) in a volume of 40 μL/25 g mouse. For injections, NDTG and NMDTG were diluted to 28.1 mg DTG/mL in endotoxin-free water and 42.3 mg MDTG/mL in HEPES buffer, respectively. Twenty-five microliters of whole blood was collected by cheek puncture into 1 mL ACN at 1, 3, 7, 14, 21, 28, 35, 42, 49, and 56 days after drug administration. Animals were humanely euthanized using isoflurane followed by cervical dislocation at 28 and 56 days, and tissues (spleen, liver, lymph nodes, lungs, kidneys, gut-associated lymphoid tissue (GALT), and brain) were collected for drug quantitation. Drug concentration in whole blood and tissues was determined by UPLC-MS/MS using a Waters Acquity UPLC- Xevo TQ-S micro mass spectrometry system (Milford, MA). For blood analysis, 10 μL of internal standard (IS) solution was added to each sample. DTG-d3 and myristoylated cabotegravir (MCAB) were used as IS for DTG and MDTG analysis, respectively. The final IS concentration was 50 ng/mL after reconstitution. Samples were then vortexed and centrifuged at $17,000 \times g$ for 10 min at 4 °C. Supernatants were dried using speed vacuum, reconstituted in 100 μL 50% (v/v) ACN in LC-MS-grade water, and 10 μL subsequently injected for DTG and MDTG analysis. Standard curves were prepared in blank mouse blood in the range of 0.2–2000 ng/mL of the corresponding drug. For tissue analysis, 50–200 mg of each sample was homogenized in four volumes of 90% (v/v) ACN in LC-MS-grade water using a Qiagen TissueLyzer II (Valencia, CA). Subsequently, 80 μL of ACN, 10 μL of 50% ACN (v/v) in LC-MS-grade water, and 10 μL IS was added to 100 μL of tissue homogenate. Standard curve samples were prepared the same way as the study samples, except for using 10 μL of 10× analyte spiking solution that results in a final concentration range of 0.5–2000 ng/mL. Chromatographic separation of 10 μL sample injections was achieved with an ACQUITY UPLC-BEH Shield RP18 column (1.7 μm, 2.1 mm × 100 mm) using a 10-min gradient of mobile phase A (7.5 mM ammonium formate in Optima-grade water adjusted to pH 3 using formic acid) and mobile phase B (100% Optima-grade ACN) at a flow rate of 0.25 mL/min. The initial mobile phase composition was 40% B for the first 3 min at which time it was increased to 86% B over 30 s and held constant for 5 min. Mobile phase B was then reset to 40% over 15 s and held for 1.25 min for equilibration. MDTG was quantified using a 7-min gradient of the same mobile phases at a flow rate of 0.35 mL/min. The initial mobile phase composition was 80% B for the first 4.5 min at which time it was increased to 95% B over 15 s and held constant for 1 min. Mobile phase B was then reset to 80% over 15 s and held for 1 min. DTG and MDTG were detected at cone voltages of 10 V and 16 V and collision energies of 25 V and 44 V, respectively. Multiple reaction monitoring (MRM) transitions used for DTG, MDTG, DTG-d3, and MCAB were 420.075 > 277.124, 630.2 > 420.067, 422.841 > 129.999, and 616.277 > 406.094 $m/z$, respectively. Spectra were analyzed and quantified by MassLynx software version 4.1. Quantitation was based upon drug peak area to internal standard peak area ratios.

**In vivo measures of HIV-1 restriction and protection**. NOD/scid-IL-2Rγc^null (NSG) mice (6–7 weeks of age; Jackson Labs, Bar Harbor, ME) were administered a single IM injection in a volume of 40 μL/25 g mouse of NDTG or NMDTG at a concentration of 45 mg/kg DTG-eq. on day 0 for measures of relative viral restriction. Mice were then reconstituted with $25 \times 10^6$ human peripheral blood lymphocytes (hu-PBL) per mouse 4 or 19 days post drug treatment by IP injection. Next, mice were challenged 10 days post hu-PBL reconstitution with HIV-1$_{ADA}$ at 14 or 29 days post drug treatment by IP injection at a TCID$_{50}$ of $2 \times 10^4$ per mouse. Blood samples were collected one day before HIV-1 infection for determination of T cell populations by flow cytometry, including: CD45+, CD3+, CD4+, CD8+, and CD4:CD8+ ratios, and regularly after treatment for determination of drug levels by UPLC-MS/MS. Animals were humanely euthanized 10 days post HIV-1 challenge and blood, plasma, and tissues were collected. Plasma was saved for viral load (HIV-1 RNA) quantitation using the Roche Amplicor and Taqman-48 system [HIV-1 kit V2.0 according to the manufacturer's instructions (Indianapolis, IN)]. Blood and tissue drug concentrations were determined by UPLC-MS/MS as described above. Tissue immunohistochemistry was performed for HIV-1p24 antigen and HLA-DP/DQ/DR[64–66]. Briefly, tissues were fixed in 4% PFA overnight and embedded in paraffin. Five-micron thick tissue sections were cut and mounted

on glass slides, which were stained for HIV-1p24 or HLA-DP/DQ/DR and developed with DAB. The nuclei were counterstained with Mayer's hematoxylin and visualized with a ×20 objective on a Nikon Eclipse E800 microscope (Melville, NY) with Nuance EX multispectral imaging system (PerkinElmer, Hopkinton, MA). The number of HIV-1p24+ and HLA-DP/DQ/DR+ cells per section were counted and expressed as percentage of HIV-1p24+ cells/HLA-DP/DQ/DR+ cells per section. For viral protection tests, NSG mice were reconstituted at birth with CD34+ HSC isolated from umbilical cord blood[67]. Reconstituted mice were administered a single IM dose of NMDTG at a concentration of 45 mg/kg DTG-eq. at 22 weeks of age. Mice were challenged two weeks post drug treatment with $2 \times 10^4$ TCID$_{50}$ of HIV-1$_{ADA}$ by IP injection. Blood was collected by cheek puncture into EDTA-blood collection tubes three weeks post HIV-1 challenge; and blood and plasma were collected to determine drug levels and viral load, respectively. Plasma was used for quantitative viral load (measurements of HIV-1 RNA) using the Roche Amplicor and Taqman-48 system. Blood and tissue drug concentrations were determined by UPLC-MS/MS as described above. For the detection of viral RNA copies, RNAscope was performed (Advanced Cell Diagnostics, Hayward, CA). A channel 1 anti-sense HIV-1 Clade B target probe, which contains 78 probe pairs targeting base pairs 854–8291 of HIV-1, was used in the single-plex chromogenic assay. Briefly, 5 μm thick de-paraffinized and dehydrated formalin-fixed paraffin-embedded (FFPE) spleen and lymph node sections were pretreated with hydrogen peroxide at room temperature for 10 min, boiling citrate buffer for 8 min then protease IV at 40 °C for 15 min in a HybEZ hybridization oven. Hybridization with target probe, pre-amplification, amplification, and chromogenic detection using DAB was carried out as per manufacturer's instructions in HybEZ oven at 40 °C. Positive expression was indicated by the presence of brown dots in the infected cells and scored using the criteria: 0 = no staining or <1 dot/10 cells, 1 = 1–3 dots/cell, 2 = 4–9 dots/cell with no, or very few, dot clusters, 3 = 10–15 dots/cell and <10% dots are in clusters, 4 = >15 dots/cell and >10% dots are in clusters. Scoring was performed at ×20 magnification. For tissue DNA and RNA isolation, each sample was homogenized using a Qiagen TissueLyzer II (Valencia, CA) and Qiagen AllPrep DNA/RNA Mini Kit (Hilden, Germany) utilized as per manufacturer's instructions. cDNA was produced from RNA using ThermoScientific Verso cDNA Synthesis Kit (Vilnius, Lithuania) as per manufacturer's instructions. Cell-associated HIV-1 RNA and DNA were quantified by semi-nested real-time PCR and confirmed by droplet digital PCR[45,68].

**Statistics**. For all studies, data were analyzed using GraphPad Prism 7.0 software (La Jolla, CA) and presented as the mean ± the standard error of the mean (SEM). Experiments were performed using a minimum of three biologically distinct replicates. Sample sizes were not based on power analyses. For comparisons of two groups, Student's $t$ test (two-tailed) was used. Tissue drug levels, HIV-1 RT activity, HIV-1p24 staining, T cell populations, viral RNA and DNA, and viral load were analyzed by one-way ANOVA with Bonferroni correction for multiple-comparisons. For studies with multiple time points, two-way factorial ANOVA and Bonferroni's post hoc tests for multiple comparisons were performed. Animal studies included a minimum of six animals per group unless otherwise noted. Extreme outliers beyond the 99% confidence interval of the mean and 3-fold greater than the SEM were excluded. Significant differences were determined at $P < 0.05$.

**Study approval**. All experimental protocols involving the use of laboratory animals were approved by the UNMC Institutional Animal Care and Use Committee (IACUC) ensuring the ethical care and use of laboratory animals in experimental research. All animal studies were performed in compliance with UNMC institutional policies and NIH guidelines for laboratory animal housing and care. Human blood cells were isolated by leukapheresis from HIV-1/2 and hepatitis seronegative donors and were deemed exempt from approval by the Institutional Review Board (IRB) of UNMC. Human CD34+ hematopoietic stem cells were isolated from umbilical cord blood and are exempt from UNMC IRB approval.

**Data availability**. Data are available from the corresponding author upon reasonable request. https://doi.org/10.6084/m9.figshare.5728299, https://doi.org/10.6084/m9.figshare.5728293, https://doi.org/10.6084/m9.figshare.5728295, https://doi.org/10.6084/m9.figshare.5727086.

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

## Acknowledgements

We wish to thank the University of Nebraska Medical Center Cores: Electron Microscopy (Tom Bargar and Nicholas Conoan), Elutriation and Cell Separation (Myhanh Che and Na Ly), Flow Cytometry (Victoria Smith), and Comparative Medicine for technical assistance. We thank Dr. Shah Valloppilly of the University of Nebraska-Lincoln Nebraska Center for Materials and Nanoscience X-Ray Structural Characterization Facility for support in characterizing the modified antiretroviral drug(s) used in this study. Finally, we thank Bhagya Laxmi Dyavar Shetty, Celina Prince, and Sruthi Sravanam for technical assistance in drug level analysis, tissue sectioning, and immunohistopathology. The research was supported by ViiV Healthcare; the University of Nebraska Foundation (donations from the Carol Swarts, M.D. Emerging Neuroscience Research Laboratory, the Margaret R. Larson Professorship, and the Frances and Louie Blumkin, and Harriet Singer Endowment); the Vice Chancellor for Research Office, University of Nebraska Medical Center, for core facility developments; and National Institutes of Health grants, P01 DA028555, R01 NS36126, P01 NS31492, 2R01 NS034239, P01 MH64570, 3P30 MH062261, P30 AI078498, 1R24 OD018546, and R01 AG043540.

## Author contributions

B. S.: Design and execution of most experiments, data acquisition, data analysis and interpretation, writing of manuscript; A. N. B.: design and execution of experiments, supervision of experiments, data acquisition, data interpretation; P. K. D.: design and execution of pharmacodynamic experiments, supervision of experiments, data analysis and interpretation; B. B.: design and execution of some experiments, data acquisition, data analysis and interpretation; T. K.: data acquisition, data analysis; S.M.: data acquisition, data analysis and interpretation; H. S.: data acquisition, data analysis and interpretation; G. D. K.: study design, data interpretation; L. Y. P.: study design; S. G.: study design; J. M.: study design, design of animal experiments, supervision of experiments, data analysis; N. G.: data acquisition, data analysis; Y. A.: study design, data interpretation; B. E.: conceived project, study design, design of synthesis, and formulation experiments, supervision of experiments, data analysis, and interpretation; H. E. G.: conceived project, design of experiments, data interpretation, writing of manuscript, and funding acquisition. All authors critically evaluated the manuscript prior to submission.

## Additional information

**Competing interests:** The authors declare no competing financial interests.

