## [Peer Review File · Nature Communications]

Reviewers' comments:

Reviewer #1 (Remarks to the Author):

In the manuscript "Creation of long-acting nanoformulated dolutegravir" authors synthesized the myristoylated dolutegravir (NMDTG) and encapsulated it to poloxamer nanoparticles. This modification led to easy uptake and retention in monocytes/macrophages. After intramuscular administration of BALB/cJ mice (single dose of 45mg DTG/kg), concentration of DTG was above or at IC90 (64ng/ml) for 56 days in plasma and 28 days in selected tissues. Authors also used the DTG nanoparticles to protect adult peripheral blood lymphocyte reconstituted mice from parenteral HIV challenge. In previous publications by the same group, myristoylation of the prodrugs and encapsulation to nanoparticles also resulted in an uptake and retention by macrophages (3TC, abacavir, citations 10 and 11 of the manuscript). In this manuscript, the same approach with DTG resulted in longer retention of the drug in macrophages in vitro and more favorable PK in vivo than before. It is essential that authors validate favorable properties of NMDTG shown in vitro also in some in vivo model. This should include models able to show extended suppression of viral replication compared to original drug (to validate extended retention of the drug in target cells), or significant reduction of HIV reservoirs (to validate better targeting of the drug to reservoirs). In this manuscript, authors attempted to validate the system for pre-exposure prophylaxis in human peripheral blood lymphocytes humanized mice.

Major comments:

- Effective pre-exposure prophylaxis, must prevent transmission of HIV infection with no detectable HIV replication and viral DNA integration in protected individuals. It was previously shown in various animal models that HIV integrase inhibitors can be used for prevention strategies (e.g. long-acting nanosuspension of cabotegravir). Authors claim that NMDTG can protect from HIV challenge 2 weeks after drug administration. However, in Figure 6b there seems to be some residual viral replication above the level of detection in some mice. To consider a mouse protected from HIV transmission, it is necessary to show that 1) no viral replication in peripheral blood/plasma and tissues occurred in the protected mouse, 2) no cell-associated or integrated viral DNA in multiple tissue is observed. It should be clearly shown how many mice in each group met those requirements. Mice were killed and tissues harvested 10 days after HIV challenge. This is short period of time for validation of the protection and longer time should be considered.
- HIV-p24 was measured using IHC in Figure 6 b,d,f,g. This approach is not very sensitive. To evaluate HIV replication in tissues, cell-associated viral RNA measured by qPCR should be used instead.
- It is not clear why a model with reconstituted lymphocytes was used. Most data in the manuscript are generated using human monocytes/macrophages, but the in vivo model has only human lymphocytes. It is difficult to connect in vitro data shown in the manuscript with the in vivo experiment. It is also impossible to validate suggested mechanism (retention of particles in macrophages) as no data from NSG (genetic background of the model) macrophages are shown. A model with at least partially reconstituted human monocyte/macrophage compartment should be used for these purposes.
- Figure 4 a,b: Please explain why HIV-1 control has lower activity than treatment with DTG

and NDTG (250% and 150%, respectively). What is the effect of tested drugs on viral replication in macrophages?

- Pg. 18 lines 339-341: In these lines authors are explaining less than one month protection by MNDTG in the mouse model by "large number of activated lymphocytes, timing of human cell reconstitution, larger viral challenge, route of viral challenge, and monotherapy approach". Although authors did not provided convincing proof that protection was achieved 2 weeks post-administration, it is also important to explain, how activated lymphocytes were measured? Suppl. Figure 2 does not show any activation marker. How timing of reconstitution in this model can be changed to improve outcome? Is it possible to use lower viral challenge and different route of transmission in this model? Did authors identify any mutations in breakthrough infection? Authors should also consider a possibility that amount of DTG in target tissues was not high enough to protect mice from HIV challenge and repeat the experiment with higher dose.

Minor comments:

- The abstract has several incorrect statements: 1) in the line 32 the statement "NMDTG was optimized" should be change to "characterized" as no optimization is shown in the result section. 2) in the line 35- tissues concentration of DTG is above IC90 only up to 28 days not 56 days. 3)line 36-as in details explained in comments above, model/experiment does not convincingly show protection from HIV transmission. 4) line 38- if author will show that no replication and viral DNA integration occur at 2 weeks post-administration, statement "greater than two weeks" should be change to simple "two weeks" as the later point (4 weeks) does not show any protection.
- Pg7; line 128: Please explain here what P407 stands for.
- Pg.7; lines 135-137: NMDTG shows changes in size and zeta potential at 25C. How this influences the uptake by macrophages and other properties of the nanoparticles?
- Pg. 9; lines 168: statement "NMDTG was also retained within MDM for up to 30 days" is incorrect. In the Figure 3d, the observation period was 30 days, NMDTG was found in cells at day 20 but not at day 30.
- Figure 3 f: Please indicate what orange and green bars stands for.
- Pg 10, line 198: in the statement "NDTG vs. NDTG control" it is not clear what NDTG control is.
- What volume of the drug was injected IM to mice in PK and the protection experiment?
- Figures 6 f, g: HLA-DR is not a general marker for human cells. Why authors use it in this setting (i.e. reconstituted human lymphocytes)?
- Pg. 18; line 340 should be "activated lymphocytes"
- Pg. 18; lines 341-344: in the manuscript, there are no data suggesting retention of the particles in the reticuloendothelial system or formation of second drug reservoir beyond the injection site. This sentence should be rephrased to be consistent with observations in the manuscript.

Reviewer #2 (Remarks to the Author):

Review comments for NCOMMS-17-17675-T

“Creation of a Long-Acting Nanoformulated Dolutegravir”.

Tamio Fujiwara

This manuscript describes chemical modification of dolutegravir to myristoylated prodrug (MDTG), and poloxamer encased nanoformulation was prepared with high-pressure homogenization.

The manuscript also includes nanoformulation stability and release kinetics, nanoparticle characterization, antiretroviral efficacy, effect of macrophage-released DTG on spreading CD4+ T cell infection, mouse pharmacokinetics with IM injection, and PrEp experiments with humanized mice.

This nanoformulated modified DTG (NMDTG) showed that NMDTG was easily incorporated into macrophages and slowly released DTG from the cells and has long half life. This long-acting parenteral NMDTG potentially increases the clinical utility of already approved DTG as oral tablet. Therefore this manuscript is worth publishing.

The reviewer has some comments shown below. My most important comment/question is #6.

1. Introduction, page 3, line 50: Although referred a review article, "RASER ART" is not easily understood. Please include explanation or change wording.
2. Page 3, line 64: It says “not yet enabled a monthly or longer frequency of administration”. However, cabotegravir (a sister drug of DTG) and rilpivirine LAP Ph2b study, LATTE-2 results have already been published and Phase III Sword 1/2 study results presented at CROI this year.
3. Page 4, line 73: reversible myristoylation. It could be misunderstood that MDTG is chemically reversible, but conversion back to DTG needs esterase as shown in Fig.1.
4. Page 4, line 75-76: MDTG was rapidly taken up by MDM. Is it already known that a myristoylated drug rapidly taken up by MDM or is this a new finding in this manuscript? If this phenomenon is already known, please refer to a relevant paper.
5. Page 4, line 80-81: intrinsic anti-viral activity of DTG does not changed, so better to include in vivo antiretroviral efficacy due to improved DMPK profile.
6. Results, page 5, line 106: anti-HIV activity measured by RT activity for MDTG is 62.5 nM. However, myristoyl residue was linked to 7-hydroxyl group of DTG. This OH is important to make chelate with Mg co-factor in inhibiting integrase strand transfer activity. So, it is hard to believe MDTG has similar activity to DTG. Was there any possibility that MDTG was rapidly hydrolyzed to DTG in the assay medium or in MDM? If the authors think MDTG itself is an active inhibitor, better to do the in vitro integrase strand transfer inhibition enzyme assay to show it. If MDTG itself is active form, then to say it a prodrug may be inappropriate.
7. It is better to describe somewhere in the manuscript that how abundantly the esterase exists in assay medium, in human plasma, in cells and so on and how rapidly MDTG hydrolyzed to DTG in some in vitro condition.
8. Pharmacokinetics (PK), page 13, L247: Please include what kind of vehicle was used for IM injection?
9. Discussion, page 20: Please include some discussion for human application of NMDTG in

regards to dose and dosing volume compared to cabotegravir in Ph3.

Reviewer #3 (Remarks to the Author):

Dear Editor and Author,

The submitted manuscript by H.E. Gendelman and co-workers details significant follow-up to the group's initial AAC report around the discovery and PK characterization of a lipidated Dolutegravir prodrug delivered as a poloxamer nanoformulation. The studies detailed in the manuscript highlight a number of interesting in vitro and in vivo follow-up studies to the initial compound discovery, including rigorous physicochemical characterization, in vitro uptake and release studies in relevant cells (i.e. MDM), in vitro antiviral activities and viral spread suppression, as well as PK and PrEP studies. Together, the data are quite well presented and paint a detailed picture of the benefits not only of long-acting parenteral administration over other routes, but also, importantly, the significantly differentiated profile of NMDTG (nanoformulated prodrug) over NDTG. The results are impressive, in my opinion. The authors set up the data for future discussions on progress of this and related formulations to further drive LAP potential as way to improve adherence and effectiveness of ART regimens.

Rather than providing line-by-line comments/edits, a detailed comment/review of the content of the manuscript is included in "PDF Comment" format in the attached file. Please review this and follow up with any questions or comments that might arise, including responses to questions raised.

Overall, a very nice piece of work. Thank you for the opportunity to review.

Sincerely,
Reviewer

Summary of Comments from Reviewer 3

Page: 1

Author: redacted
long-acting

Page: 2

Author: redacted
I recommend softening this language as there are a number of clinical assets that match the viral resistance profile of DTG.

Author: redacted:
Lymphocytes

Page: 3

Author: redacted: Please clarify what you mean by "interpatient PK profiles are operative".

Author: redacted: I do not believe this is accurate. There are examples from the clinic (not market) of >QMo dosing. Please review your references.

Page: 4

Author: redacted:
"effectiveness" is not the correct word choice here. The information you present speak to efficacy, but not effectiveness. I recommend rewording.

Author: redacted
clarify that this is in vitro

Author: redacted
Efficacy or potency? Elaborate a bit here. How are you measuring your potency. What is driving this improved potency?

Author: redacted
long-acting

Author: redacted
In vitro as commented on above or in vivo? Clarify.

Page: 5

Author: redacted
I recommend at least one paragraph talking about your discovery of this particular promoiety as being optimal. The author does not provide any discussion of SAR. While presented elsewhere, it warrants mention here - why an ester? why this particular fatty acid? etc...

Author: redacted
These latter two are not techniques but rather measurements. I would separate them from your analytical descriptors.

Author: redacted
The detailed NMR and IR analysis here is likely not needed. Simply report spectral data in an experimental section.

Author: redacted
I am unclear as to the reason you report the myristoyl chloride IR data.

Author: redacted
Fatty acid prodrugs are notoriously difficult to crystallize. I recommend providing a bit more detail on how you accessed the crystalline form.

Author: redacted
What do you mean by HIV-1 RT activity? What does RT stand for here? Your activity descriptor is unclear.

Author: redacted
This latter statement is a bit confusing. The potency observed for MDTG necessitates the loss of the promoiety in your assay. DTG without access to the -OH will not be potent. Please comment on this in your description of potency. Were you to test MDTG in a biochemical assay in which the promoiety is not cleaved, you would not see potency.

Page: 6

Author: redacted

Are these hashes or wedges? They look like wedges. If so, your stereochemistry is not correct. Review and correct the structure.

Author: redacted

I recommend cleaning up your structures a bit. the amide bond is not visible as drawn. Also, your hash/wedge depictions are difficult to see clearly. Narrower bond widths would help.

Author: redacted

Can you comment on chemical stability? This ester is likely cleaved chemically as well. What have you done to study the mechanism of bioconversion?

Author: redacted

I assume this is a cell-based assay? Please clarify.

Page: 7

Author: redacted

How were the crystals generated. This is not yet described, but a critical outstanding question.

Author: redacted

Your number in the text is 82.8. Be consistent.

Author: redacted

Confirms that this is a cell-based assay.

Author: redacted

Again, please review my earlier comments about this potency. Text should be clarified.

Page: 8

Author: redacted

General comment, ranges and minus signs should be indicated as n-dashes, not dashes.

Author: redacted

Clarify what P407 is. Poloxamer 407.

Author: redacted

I believe this is the first time you use this acronym. Define it here first.

Page: 11

Author: redacted

comma, not semicolon

Author: redacted

Can you elaborate on this a little?

Author: redacted

Indicate here that this is an in vitro system, and please provide a little more detail around your methods.

Author:

...concentrations increased...

Author: redacted

in vitro

Author: redacted

Data

Author: redacted

MDMs

Author: redacted

Provide more detail on the effect observed.

Author: redacted

MDMs

Author: redacted

Again, please provide a little more context here in the main body of the manuscript.

Author: redacted

How was phagocytic function measured?

Page: 12

Author: redacted

What do these red lines represent? Can you provide a scale to assess size?

Page: 13

Author: redacted

rod-shaped

Author: redacted

Indicate this on the graphic.

Author: redacted

rod-shaped

Page: 14

Author: redacted

What is RT?

Author: redacted

An

Author: redacted

replace 'were determined' with 'was carried out'

Author: redacted

8-hour

Author: redacted

MDMs

Author: redacted

'activities were only operative' is awkward word choice. Consider rewording.

Page: 15

Author: redacted

Your are displayed in a different order from that which you discuss them in the text. Consider reordering to make the flow more logical.

Page: 17

Author: redacted

Good section. Clear and well-written.

Author: redacted

This is an unusual dose unit for individuals not used to thinking this way. Please clarify how you calculate/measure this. Most readers will be more used to mg/kg units.

Author: redacted

remove their

Author: redacted

Over the course of the study? Did animals not gain weight or did they not have abnormal weight gain. Animals should have gained weight over 8 weeks. Please clarify. Your graph makes this clear, but your text does not.

Author: redacted

Should this be "effective" or "apparent" half-life? The intrinsic half-life of DTG does not change. Your PK is being driven by CL and release from the IM depot, not half-life. Be sure this distinction is clear.

Author: redacted

Same comment.

Author: redacted
write out this acronym

Page: 20

Author: redacted
Same comments around units.

Author: redacted
Can you comment or provide references around this model? Mice are generally much less reliable models for HIV compared to monkeys (SHIV). This model warrants a bit more discussion as to its validity.

Author: redacted
Good discussion.

Page: 24

Author: redacted
Currently

Author: redacted
could or should

Author: redacted
Be consistent with the rest of your paper and refer to these as MDM

Author: redacted
"screening" is not the right word here. consider "evaluation"

Author: redacted
Apparent

Author: redacted
I would reword "PrEP tests" to "evaluation of pre-exposure prophylactic potential"

Author: redacted
Good references. Include in earlier discussion.

Author: redacted
MDM

Page: 25

Author: redacted
Isn't the fundamental property that underlies all of these the slow release of prodrug/drug into systemic circulation? All the properties you discuss contribute to this.

Author: redacted
"target" might not be the correct word here. Consider "enable enhanced delivery." Target suggests that you are selectively delivering drug to a specific location, and you are not.

Author: redacted
This is a controversial topic, and I am not confident that you have indicated that your delivery technology truly targets viral reservoirs. Your LN data are intriguing, but there are caveats associated with it. Consider qualifying this a bit more.

Author: redacted
MDM

Author: redacted
MDM

Author: redacted
"Trojan Horses"

Author: redacted
This discussion and these references provide a bit more context around my comment on line 361 above. Consider changing the order in which you present the content in this paragraph.

Author: redacted
Reiterating my comment around the term "half-life" here. You measure systemic levels of drug and that intrinsic half-life does not change.

Page: 26

Author: redacted
Word choice. Re-emerging?

Author: redacted

This is another big topic and one that is a bit prematurely brought up in this discussion. Consider removing content associated with cure from your discussion.

Page: 27

Author: redacted
Same comment as above. Discussion of cure and functional cure does not belong in this manuscript, in my opinion.

Page: 28

Author: redacted
I have not reviewed the formatting of the references. I assume journal editors will do this.

Page: 34

Author: redacted
n-dash

Author: redacted

You should include an actual synthetic experimental (concentrations, equivalents, times, etc.) and corresponding characterization data - ¹H NMR, ¹³C NMR, IR, optical rotation, etc. Your section is too much of a discussion than a technical procedure.

Page: 35

Author: redacted
You mention the instruments but do not show the data. You should indicate the latter. See any chemistry journal for how to do this in a technically accurate manner.

Author: redacted

What did you do to optimize the crystalline phase? This is not discussed earlier, and your experimental here suggests that you simply moved forward with the phase that came out of diethyl ether. With the phase critical to formulation, please provide some discussion on what was done to optimize.

Page: 36

Author: redacted
10-microliter

Page: 37

Author: redacted
How did you assess this. What was your collection and lysis procedure?

Page: 38

Author: redacted
n-dash

Page: 40

Author: redacted
40 uL/25 g (no dash)

Author: redacted
What muscle? Quad?

Author: redacted
To account for the promoiety.

Author: redacted
What was the process for euthanizing?

Author: redacted
The final

Author: redacted
no spaces with dash.

Author: redacted
n-dash

Page: 41

Author: redacted
10-uL

Author: redacted
n-dash

Author: redacted
what muscle? Quad?

Page: 42

Author: redacted
What was your process for euthanizing?

Author: redacted Subject: Sticky Note Date: 8/25/2017 11:32:27 AM -04'00'
5-um

Page: 45

Author: redacted
This assay is not described in your Methods section.

Author: redacted
n-dash

Author: redacted
n-dash

Author: redacted Subject: Sticky Note Date: 8/25/2017 11:34:52 AM -04'00'
Assay not described in Methods section.

Page: 48

Author: redacted
Ensure that your PK units selection are the correct conventions for the journal to which you are submitting.

Page: 49

Author: redacted
Why is this page left blank?

Re: NCOMMS-17-17675-T, "Creation of a Long-Acting Nanoformulated Dolutegravir".

Reviewer 1

Overall. *"In the manuscript "Creation of long-acting nanoformulated dolutegravir" authors synthesized the myristoylated dolutegravir (NMDTG) and encapsulated it to poloxamer nanoparticles. This modification led to easy uptake and retention in monocytes/macrophages. After intramuscular administration of BALB/cJ mice (single dose of 45mg DTG/kg), concentration of DTG was above or at IC90 (64ng/ml) for 56 days in plasma and 28 days in selected tissues. Authors also used the DTG nanoparticles to protect adult peripheral blood lymphocyte reconstituted mice from parenteral HIV challenge. In previous publications by the same group, myristoylation of the prodrugs and encapsulation to nanoparticles also resulted in an uptake and retention by macrophages (3TC, abacavir, citations 10 and 11 of the manuscript). In this manuscript, the same approach with DTG resulted in longer retention of the drug in macrophages in vitro and more favorable PK in vivo than before."*

Response: Agree.

Major comments

Point 1. *"It is essential that authors validate favorable properties of NMDTG shown in vitro also in some in vivo model. This should include models able to show extended suppression of viral replication compared to original drug (to validate extended retention of the drug in target cells), or significant reduction of HIV reservoirs (to validate better targeting of the drug to reservoirs). In*

this manuscript, authors attempted to validate the system for pre-exposure prophylaxis in human peripheral blood lymphocytes humanized mice.”

Response: Agreed. The new experiments now included were designed to cross validate the initial reported properties of NMDTG. These were performed utilizing a different humanized mouse model [NOD/SCID/IL2R γ ^{-/-} (NSG)] of HIV/AIDS. In these experiments mice were reconstituted with human cord blood CD34+ hematopoietic stem cells (HSC). The NSG-HSC mouse model better reflects the human condition as the reconstitution contains monocyte-macrophages as well as CD4+ and CD8+ T lymphocytes. All cell types are notably not activated prior to viral challenge (Response Fig. 1). Thus, the model better reflects the *in vitro* experiments developed in this report and demonstrates the interactions between macrophages and LASER ART in protection against viral challenge. To this end, two additional pre-exposure prophylaxis (PrEP) experiments were performed. The first used NMDTG alone (monotherapy) (Manuscript Fig. 6), as was suggested in review and the second used a combination of NMDTG with nanoformulated rilpivirine (NRPV) (dual therapy) for comparison (Response Fig. 2). Drug concentrations were 45 mg/kg DTG-equivalents and 45 mg/kg RPV administered as a single and sequential dose, respectively. Both experiments showed extended suppression of viral replication.

Point 2. *“Effective pre-exposure prophylaxis must prevent transmission of HIV infection with no detectable HIV replication and viral DNA integration in protected individuals. It was previously shown in various animal models that HIV integrase inhibitors can be used for prevention strategies (e.g. long-acting nanosuspension of cabotegravir). Authors claim that NMDTG can protect from HIV challenge 2 weeks after drug administration. However, in Figure 6b there seems to be some residual viral replication above the level of detection in some mice. To considered a mouse protected from HIV transmission, it is necessary to show that 1) no viral replication in peripheral blood/plasma and tissues occurred in the protected mouse, 2) no cell-associated or integrated viral DNA in multiple tissue is observed. It should be clearly shown how many mice in each group met those requirements. Mice were killed and tissues harvested 10 days after HIV challenge. This is short period of time for validation of the protection and longer time should be considered.”*

Response: Agreed. The adult peripheral blood lymphocyte (PBL) reconstituted NSG mice showed partial viral breakthrough, based, in part, due to a generalized activated immune system proven highly susceptible to HIV-1 infection (Response Fig. 1). The PBL reconstituted mice were infected after just 3 days while CD34+ humanized mice were infected after 14 days. These differences highlight the robust activation and susceptibility of the PBL model to HIV-1 infection. We also agree that the results require validation with a more physiologically relevant model that contains monocyte-macrophage and would require complete protection for experimental validation. The limitations and strengths of both models are better detailed in the discussion of the new manuscript and the text provided below. The reviewer is correct, and we thus accept the limitations of this first model that is now used simply as a screen for relative differences in viral restriction. To all ends we repeated the experiment using humanized mice where both CD4+ and CD8+ T cells and monocyte-macrophages were present. In the monotherapy

experiment, mice were challenged 2 weeks after drug administration, whereas in the dual therapy experiment, mice were challenged 4 weeks post-NMDTG administration (2 weeks post-NRPV administration). In monotherapy, mice were protected up to 3 weeks after viral challenge. Notably in dual therapy, mice were protected up to 6 weeks post-viral challenge. Such viral protection was confirmed with plasma viral load, RNAscope, and semi-nested PCR and exhaustively detailed in the “new” Figure 6. The amended illustrations below are meant only as a reference to help guide the reviewer and these data will be placed in other manuscripts at a relevant future time.

Point 3. *“HIV-p24 was measured using IHC in Figure 6 b,d,f,g. This approach is not very sensitive. To evaluate HIV replication in tissues, cell-associated viral RNA measured by qPCR should be used instead.”*

Response: Agreed. HIV-1p24 measurement was done using IHC for preliminary evaluation. In new experiments, data were obtained with RNAscope and semi-nested PCR, and are now included in the revised text.

Point 4. *“It is not clear why a model with reconstituted lymphocytes was used. Most data in the manuscript are generated using human monocytes/macrophages, but the in vivo model has only human lymphocytes. It is difficult to connect in vitro data shown in the manuscript with the in vivo experiment. It is also impossible to validate suggested mechanism (retention of particles in macrophages) as no data from NSG (genetic background of the model) macrophages are shown. A model with at least partially reconstituted human monocyte/macrophage compartment should be used for these purposes.”*

Response: Understood. The original PBL model was used as an initial screening tool to test the pharmacodynamic efficacy of NDTG and NMDTG and to demonstrate that NMDTG was effective in lymphocytes as well as monocyte-macrophages. As stated above, however, we accept the concerns outlined by the reviewer and the need to cross validate and extend the observations in a second, complete humanized mouse model. Notably, the caveats and limitations of the prior experiment using the PBL model are fully detailed. The CD34+ humanized mice added are fully explained as to their genetic background and the presence of macrophages as well as lymphocytes. This model more closely reflects what is operative in humans and is now detailed with full illustrated data sets (Manuscript Fig. 6). The original study in PBL reconstituted mice has now been moved to the supplement (Supplementary Fig. 2).

Point 5. *“Figure 4 a,b: Please explain why HIV-1 control has lower activity than treatment with DTG and NDTG (250% and 150%, respectively). What is the effect of tested drugs on viral replication in macrophages?”*

Response: DTG or NDTG showed protection in early time points after HIV-1 challenge (up to 4 hours), causing partial infection of plated cells. Due to poor intracellular retention properties, with time drug was washed out and protected cells became vulnerable to infection. Whereas, in the infected group, all cells were susceptible to virus and demonstrable infection occurred at

early time points with limited cytopathicity. This helps explain higher RT activity in later time points in the DTG or NDTG treated groups compared to HIV-1 infected control, as greater number of cells are present in these cultures as compared to cells not treated with DTG where significant cell death was readily observed.

Point 6. *“Pg. 18 lines 339-341: In these lines authors are explaining less than one month protection by MNDTG in the mouse model by “large number of activated lymphocytes, timing of human cell reconstitution, larger viral challenge, route of viral challenge, and monotherapy approach”. Although authors did not provided convincing proof that protection was achieved 2 weeks post-administration, it is also important to explain, how activated lymphocytes were measured? Suppl. Figure 2 does not show any activation marker. How timing of reconstitution in this model can be changed to improve outcome? Is it possible to use lower viral challenge and different route of transmission in this model? Did authors identify any mutations in breakthrough infection? Authors should also consider a possibility that amount of DTG in target tissues was not high enough to protect mice from HIV challenge and repeat the experiment with higher dose.”*

Response: Thank you. These questions are addressed in the revised discussion and the points are completely vetted.

Minor comments

Point 7. *“In the line 32 the statement “NMDTG was optimized” should by change to “characterized” as no optimization is shown in the result section.”*

Response: Agreed. Suggested change is made in the revised text.

Point 8. *“In the line 35- tissues concentration of DTG is above IC90 only up to 28 days not 56 days.”*

Response: Yes, you are correct and this error is fixed. Thank you.

Point 9. *“Line 36-as in details explained in comments above, model/experiment does not convincingly show protection from HIV transmission.”*

Response: Agreed and expanded appropriately as suggested.

Point 10. *“line 38- if author will show that no replication and viral DNA integration occur at 2 weeks post-administration, statement “greater than two weeks” should be change to simple “two weeks” as the later point (4 weeks) does not show any protection.”*

Response: Agreed and corrected through the performance of additional confirmatory experiments as mentioned above.

Point 11. *“Pg7; line 128: Please explain here what P407 stands for.”*

Response: P407 stands for poloxamer 407 (polymer used to encapsulate the prodrug and native drug nanocrystals). This explanation is now provided in the text.

Point 12. *“Pg.7; lines 135-137: NMDTG shows changes in size and zeta potential at 25C. How this influences the uptake by macrophages and other properties of the nanoparticles?”*

Response: Our group and others have extensively reported on nanoparticle uptake and retention kinetics, cell viability and efficacy of the nanoformulated drugs. According to these studies, rod shaped nanoparticles are easily recognized and internalized by macrophages (Nanomedicine (Lond). 2011 Aug; 6(6): 975-94; *PloS one* 5(4), e10051 (2010).

Point 13. *“Pg. 9; lines 168: statement “NMDTG was also retained within MDM for up to 30 days” is incorrect. In the Figure 3d, the observation period was 30 days, NMDTG was found in cells at day 20 but not at day 30.”*

Response: NMDTG was in fact detectable at day 30. The drug concentration was 31 ng/10⁶ cells.

Point 14. *“Figure 3 f: Please indicate what orange and green bars stands for.”*

Response: The figure was corrected accordingly. The orange and green bars are now fully explained.

Point 15. *“Pg. 10, line 198: in the statement “NDTG vs. NDTG control” it is not clear what NDTG control is.”*

Response: Understood and explained.

Point 16. *“What volume of the drug was injected IM to mice in PK and the protection experiment?”*

Response: The injection volume was 40- μ L/25 g mouse in both PK and protection experiments. Text is updated accordingly in the methods section.

Point 17. *“Figures 6 f, g: HLA-DR is not a general marker for human cells. Why authors use it in this setting (i.e. reconstituted human lymphocytes)?”*

Response: Text was updated to clarify that anti-human HLA-DP/DQ/DR antibody was used as a marker for preliminary evaluation of human antigen presenting cells (macrophages, B-cells, dendritic cells) and lymphocytes.

Point 18: Pg. 18; line 340 should be “activated lymphocytes.”

Response: Text was revised accordingly.

Point 19. “Pg. 18; lines 341-344: in the manuscript, there are no data suggesting retention of the particles in the reticuloendothelial system or formation of second drug reservoir beyond the injection site. This sentence should be rephrased to be consistent with observations in the manuscript.”

Response: Our laboratory demonstrated that macrophages recognize, internalize and store nanoformulated antiretroviral drugs in endosomal compartments as well as facilitate drug dissemination into the lymphoreticular system (liver, spleen, gut and lymph nodes) and T-cells. Macrophages have large storage capacities and can bypass physiological barriers to enter sites of infection, injury and inflammation. (Methods Mol Biol. 2013; 991:47-55; Methods Mol Biol. 2013; 991:41-6; Antimicrob Agents Chemother. 2013 Jul; 57(7): 3110-20; AIDS. 2012 Nov 13; 26(17): 2135-44; J Infect Dis. 2012 Nov 15; 206(10): 1577-88; Nanomedicine (Lond). 2011 Aug; 6(6): 975-94)

Reviewer 2

Overall. “This manuscript describes chemical modification of dolutegravir to myristoylated prodrug (MDTG), and poloxamer encased nanoformulation was prepared with high-pressure homogenization. The manuscript also includes nanoformulation stability and release kinetics, nanoparticle characterization, antiretroviral efficacy, effect of macrophage-released DTG on spreading CD4+ T cell infection, mouse pharmacokinetics with IM injection, and PrEP experiments with humanized mice...This long-acting parenteral NMDTG potentially increases the clinical utility of already approved DTG as oral tablet. Therefore this manuscript is worth publishing.”

Response: Thank you.

Point 1. Introduction, page 3, line 50: Although referred a review article, “LASER ART” is not easily understood. Please include explanation or change wording.

Response: LASER ART stands for “Long Acting Slow Effective Release Anti-Retroviral Therapy”. A description is provided in the introduction of the revised text.

Point 2. Page 3, line 64: It says “not yet enabled a monthly or longer frequency of administration”. However, cabotegravir (a sister drug of DTG) and rilpivirine LAP Ph2b study, LATTE-2 results have already been published and Phase III Sword 1/2 study results presented at CROI this year.

Response: The text has been revised to reflect the main limitations of the existing nanoformulations for HIV therapy that include requirement for high doses, injection volumes and injection site reactions (Markowitz *et al. Lancet HIV* 2017; (in press)). NMDTG maximizes drug loading by keeping excipient usage at a minimum. This could potentially translate into reduced dosing volumes and decreased excipient-related adverse effects.

Point 3. Page 4, line 73: reversible myristoylation. It could be misunderstood that MDTG is chemically reversible, but conversion back to DTG needs esterase as shown in Fig.1.

Response: Thank you. The word “reversible” is now deleted from line 73.

Point 4. Page 4, line 75-76: MDTG was rapidly taken up by MDM. Is it already known that a myristoylated drug rapidly taken up by MDM or is this a new finding in this manuscript? If this phenomenon is already known, please refer to a relevant paper.

Response: Lipophiles facilitate transport of molecules across cell and tissue barriers. Additional references have been included and a relevant reference is included.

Point 5. Page 4, line 80-81: intrinsic anti-viral activity of DTG does not changed, so better to include *in vivo* antiretroviral efficacy due to improved DMPK profile.

Response: Understood and agreed. New *in vivo* viral protection experiments were performed utilizing a complete humanized mouse model of HIV/AIDS. Two additional pre-exposure prophylaxis (PrEP) experiments were performed and serve to validate the conclusions reached in this paper. The first used NMDTG alone (monotherapy) (Manuscript Fig. 6) and the second used a combination of NMDTG with nanoformulated rilpivirine (NRPV) (dual therapy) for comparison (Response Fig. 2). Drug concentrations were 45 mg/kg DTG-eq. and 45 mg/kg RPV administered as a single and sequential dose, respectively. Both experiments showed extended suppression of viral replication. Text and data are now updated.

Point 6. Results, page 5, line 106: anti-HIV activity measured by RT activity for MDTG is 62.5 nM. However, myristoyl residue was linked to 7-hydroxyl group of DTG. This OH is important to make chelate with Mg co-factor in inhibiting integrase strand transfer activity. So, it is hard to believe MDTG has similar activity to DTG. Was there any possibility that MDTG was rapidly hydrolyzed to DTG in the assay medium or in MDM? If the authors think MDTG itself is an active inhibitor, better to do the *in vitro* integrase strand transfer inhibition enzyme assay to show it. If MDTG itself is active form, then to say it a prodrug may be inappropriate.

Response: Explanations of *in vitro* and *in vivo* hydrolysis of the prodrug ester results in DTG and myristic acid are now included. Myristic acid can inhibit the activity of *N*-myristoyltransferase; a crucial enzyme that catalyzes myristoylation of several proteins involved in the life cycle of HIV (PNAS, 463(4), 988-993 (2015) and *FEBS letters* 527(1-3), 138-142 (2002)). This, in addition to increased protein binding of the prodrug, provides an explanation of the improved efficacy of the myristoylated analog. There is “no evidence” in support of MDTG

itself being an active inhibitor of HIV infection. We now provide evidence that MDTG is rapidly hydrolyzed to DTG (Supplementary Fig. 1e). This was also confirmed by measuring the rapid presence of native drug in the cell culture experiments (Manuscript Fig. 3c).

Point 7. *It is better to describe somewhere in the manuscript that how abundantly the esterase exists in assay medium, in human plasma, in cells and so on and how rapidly MDTG hydrolyzed to DTG in some in vitro condition.*

Response: Additional data set on NMDTG ex vivo cleavage profile has been included in the supplementary section (Supplementary Fig. 1e). We have also referenced a detailed review article on enzymes involved in the bioconversion of ester-based prodrugs [J. Pharm. Sci. 95, 1177–1195 (2006)].

Point 8. *Pharmacokinetics (PK), page 13, L247: Please include what kind of vehicle was used for IM injection?*

Response: For NDTG, endotoxin-free water was used as a vehicle. For NMDTG, 10 mM HEPES (4-(2-hydroxyethyl)-1-piperazineethanesulfonic acid) buffer, pH 7.8, was used. Text is updated accordingly in PK section under methods.

Point 9. *Discussion, page 20: Please include some discussion for human application of NMDTG in regards to dose and dosing volume compared to cabotegravir in Ph3.*

Response: While progress was made in extending the half-life of cabotegravir, limitations abound for the current CAB LAP injectable as they require high doses, large injection volumes and show ample evidence for injection site reactions. NMDTG maximizes drug loading by limiting the amount of excipient, which could potentially translate into reduced dosing volumes and decreased excipient-related adverse effects. Compared to CAB LAP, NMDTG has enhanced cell and tissue penetration due to improved lipophilicity of the myristoylated DTG. The change of drug depot from muscle injection site (for CAB LAP) to cells and tissues (for NMDTG) may not only minimize injection site reactions but also improve delivery and sustained release of therapeutic DTG concentrations at sites of restricted HIV-1 infection.

Reviewer 3

Overall. *“The submitted manuscript by H.E. Gendelman and co-workers details significant follow-up to the group’s initial AAC report around the discovery and PK characterization of a lipidated Dolutegravir prodrug delivered as a poloxamer nanoformulation...Together, the data are quite well presented and paint a detailed picture of the benefits not only of long-acting parenteral administration over other routes, but also, importantly, the significantly differentiated profile of NMDTG (nanoformulated prodrug) over NDTG. The results are impressive, in my opinion...Rather than providing line-by-line comments/edits, a detailed comment/review of the content of the manuscript is included in "PDF Comment" format in the attached file. Please*

review this and follow up with any questions or comments that might arise, including responses to questions raised. Overall, a very nice piece of work. Thank you for the opportunity to review.”

Response: Thank you very much. The detailed comment/review of the content of the manuscript, included as a PDF comment, was excellent and very much appreciated. We attended to each point raised in detail. Also and as per reviewer’s comments, changes are made in the text. A few of the comments are addressed in more detail below.

Point 1. *“I recommend at least one paragraph talking about your discovery of this particular promoiety as being optimal. The author does not provide any discussion of SAR. While presented elsewhere, it warrants mention here - why an ester? Why this particular fatty acid? etc...”*

Response: Esters are the most utilized prodrugs now in clinical use since they are readily hydrolyzed by esterases present in the body. While prodrug structural activity studies are not presented, a paragraph highlighting the synthesis of MDTG was added to the revised text. While *not* a new compound (the prodrug undergoes bioconversion to DTG and myristic acid) a new data set on the MDTG ester bond cleavage profile was included (Supplementary Fig. 1e). The rationale for using myristic acid was based on literature reports and our prior success in developing abacavir and lamivudine formulations that showed superior pharmacokinetics compared to native drugs.

Point 2. *“I am unclear as to the reason you report the myristoyl chloride IR data.”*

Response: The myristoyl chloride spectrum provided permits further confirmation of DTG derivatization.

Point 3. *“Fatty acid prodrugs are notoriously difficult to crystallize. I recommend providing a bit more detail on how you accessed the crystalline form. How were the crystals generated? This is not yet described, but a critical outstanding question. What did you do to optimize the crystalline phase? This is not discussed earlier, and your experimental here suggests that you simply moved forward with the phase that came out of diethyl ether. ...please provide some discussion on what was done to optimize.”*

Response: The crystalline MDTG prodrug presented is a feature of the drug after conversion into a lipophilic entity. The crystal was then encapsulated into a particle by high-pressure homogenization. We did move forward with the phase that came from the diethyl ether, as it proved critical to formulation. We now provide discussion on what was done to optimize the drug system used in this report.

Point 4. *“The potency observed for MDTG necessitates the loss of the promoiety in your assay. DTG without access to the -OH will not be potent. Please comment on this in your description”*

of potency. Were you to test MDTG in a biochemical assay in which the pro moiety is not cleaved, you would not see potency.”

Response: We agree. MDTG represents an inactive and masked form of DTG. The observed improved antiretroviral activity was not only due to intracellular and tissue accumulation of the drug but also rapid cleavage of the ester bond in the prodrug to generate active DTG. New data set demonstrating bioconversion of MDTG to DTG is now provided (Supplementary Fig. 1e).

Point 5. *“Can you comment on chemical stability? This ester is likely cleaved chemically as well. What have you done to study the mechanism of bioconversion?”*

Response: Additional data sets on NMDTG cleavage profiles are now included in the supplementary data section. We have also referenced a review article on enzymes involved in the bioconversion of ester-based prodrugs for clarity on the topic of chemical stability [J. Pharm. Sci. 95, 1177 (2006)].

Point 6. *“This is an unusual dose unit for individuals not used to thinking this way. Please clarify how you calculate/measure this. Most readers will be more used to mg/kg units.”*

Response: This dose unit was made due to the larger molecular weight of the prodrug. We wished to give equimolar amounts of DTG for both treatments (NDTG and NMDTG). This enabled the dose of 45 mg/kg DTG to be constant. Therefore, an MDTG dose of 67.6 mg/kg yielded 45 mg/kg of DTG. The units were changed to “mg/kg DTG-equivalents (eq.)” throughout the manuscript with details added to avoid confusion.

All together, the manuscript was extensively revised according to the queries raised by all three reviewers. We believe these changes impact the work quality and are quite appreciative to the reviewers and editors for their thoughtful comments. Please do not hesitate to call on me if additional questions arise.

Response Figure 1. Differences in HIV-1 infectivity in CD34+ humanized mice compared to PBL reconstituted mice. Quantification of HIV-1 RNA in CD34+ humanized NSG mice (left) and PBL reconstituted NSG mice (right) tissues at different time points after infection. Gut, spleen, lung, liver, brain, and kidney viral RNA levels were determined by semi-nested PCR and calculated as viral copies per 10^6 human CD45+ cells.

Response Figure 2. LASER ART pre-exposure prophylaxis (PrEP) in CD34+ humanized mice. CD34+ hematopoietic stem cell (HSC) reconstituted NSG mice were treated according to the scheme illustrated in (a). HIV-1-infected mice without treatment served as positive controls. (b) Plasma DTG and RPV concentrations were analyzed by UPLC-MS/MS. (c) Percent CD4+ lymphocytes were measured by flow cytometry before and after HIV-1 challenge. (d) Plasma viral load was measured at terminal euthanasia after HIV-1 challenge two-weeks post LASER ART treatment. * $P = 0.0121$. (e-g) Viral DNA copies were analyzed by semi-nested qRT-PCR. (e) Lung, (f) bone marrow, and (g) spleen HIV-1gag DNA levels are shown at terminal euthanasia. (e-g) Viral RNA copies were analyzed by qRT-PCR. (h) Lung, (i) bone marrow, and (j) spleen HIV-1gag RNA levels are shown at terminal euthanasia. (k) Representative RNAscope images performed on 10- μ m thick spleen and lymph node sections using a probe targeting HIV-1 (brown). Results are shown as the mean \pm SEM of 7 LASER ART treated and 4 control animals. Findings were analyzed by two-tailed Mann-Whitney U test.

REVIEWERS' COMMENTS:

Reviewer #1 (Remarks to the Author):

Authors addressed all my comments, most of them satisfactory. Although I did not ask for additional experiments, I appreciate that in the new version of the manuscript authors added an experiment with humanized mouse model that better reflects human nature and is more appropriate to test NMDTG.

However, several inaccuracies have to be address.

1. In the new experiment, mice were administered with NMDTG, they were exposed to HIV two weeks later and analyzed 3 weeks after HIV exposure. It means that protection of mice from HIV transmission was tested only 2 weeks after drug administration, not 3 weeks as authors are claiming in the abstract (line 37). Three weeks were just point of analysis.

2. The experimental design of the protection experiment does not include a "washout" period or other approach to distinguish scenario, where HIV transmission occurred but high concentration of drug suppresses local HIV replication and prevent systemic infection. In this case, systemic HI infection will occur only after drug concentration sufficiently decrease (for example Radzio et al, Sci Transl Med, 2015). Authors analyzed animals 3 weeks post HIV exposure. As shown in Fig 6, at that time there was still 4xIC90 DTG in plasma, very probably concentration high enough to suppress potential breakthrough replication. Current data are not able to provide evidence that NMDTG protect from HIV transmission and cannot be interpreted as a proof of protection.

However, for purpose of this manuscript, it is important to show in vivo efficacy of NMDTG, not necessary protection from HIV transmission. The experiment with PBL mouse model as well as the new experiment in figure 6 provide some evidence that the drug can be effective in vivo.

Please indicate in the paragraph describing the new experiment (lines 261-291) that the experimental design did not exclude possibility that high concentration of the drug present during analysis can suppress potential breakthrough infection.

"Complete protection" is redundant term. One can be protected from HIV at the time of exposure or not. If HIV is detected in any time post-exposure, there was no protection. In lines 262,323, 331, and 333 this term is used for HIV analysis (concentration of RNA/DNA) below limit of detection of an analytical method. It should be described like that, not as a "complete protection".

Reviewer #2 (Remarks to the Author):

The major claims of this manuscript is myristoylation of DTG AND encapsulation of MDTG into poloxisamer to make NMDTG. NMDTG is easily taken up by MDM and is stable inside MDM and this made apparent half life of DTG very long. Only MDTG nor NDTG could not achieve this property.

Previous manuscript was misunderstanding that MDTG is active. However, in the revised manuscript, hydrolysis of MDTG in biological fluid is very rapid and it was caused by

esterases. Encapsulation of MDTG into polymer protects hydrolysis of MDTG. Other comments were well responded.

Minor comments:

1. P14, L299, "physiochemical" may be "physicochemical".
2. My previous Point 3, I confirmed "reversible" was deleted. However, in page 4, L71, I noticed "reversible" still remains. If the authors intend this word for the meaning that eventually NMDTG is degraded and converted back to DTG, it is OK.

Reviewer #3 (Remarks to the Author):

Dear Author and Editor,

The re-submitted manuscript detailing the synthesis, characterization, and in vitro/in vivo evaluation of a lipid prodrug of Dolutegravir has gone a long way toward addressing nearly all of the comments and suggestions provided by reviewers in the first round. I commend the authors for the diligent and thorough follow-up. Additional data include:

- clearer details associated with API characterization (both graphical and written)
- additional pieces of in vitro data
- additional in vivo (PrEP) models to paint a clearer picture of DTG performance in mouse models despite caveats associated with these models.

These extensive additions/edits significantly strengthen the manuscript and indicate a true dedication of the author to his work. In conducting a final review, there remain some minor formatting/punctuation errors that should be corrected if possible. I have not captured these in writing, but they are primarily associated with the incorrect use of dashes (dash, n-dash, m-dash) throughout the manuscript. I would recommend the author review these in detail and correct appropriately.

Otherwise, thank you for re-submitting a well re-written manuscript. Congratulations on a very nice piece of work!

Sincerely,
Reviewer

Response to reviewers' comments:

Reviewer 1

Overall. *Authors addressed all my comments, most of them satisfactory. Although I did not ask for additional experiments, I appreciate that in the new version of the manuscript authors added an experiment with humanized mouse model that better reflects human nature and is more appropriate to test NMDTG.*

Response: Thank you.

Point 1. *In the new experiment, mice were administered with NMDTG, they were exposed to HIV two weeks later and analyzed 3 weeks after HIV exposure. It means that protection of mice from HIV transmission was tested only 2 weeks after drug administration, not 3 weeks as authors are claiming in the abstract (line 37). Three weeks were just point of analysis.*

Response: Understood. We corrected the time of testing and analysis. The abstract now reads that HIV transmission was tested 2 weeks after drug administration and three weeks after drug exposure.

Point 2. *The experimental design of the protection experiment does not include a “washout” period or other approach to distinguish scenario, where HIV transmission occurred but high concentration of drug suppresses local HIV replication and prevent systemic infection. In this case, systemic HI infection will occur only after drug concentration sufficiently decrease (for example Radzio et al, Sci Transl Med, 2015). Authors analyzed animals 3 weeks post HIV exposure. As shown in Fig 6, at that time there was still 4xIC90 DTG in plasma, very probably concentration high enough to suppress potential breakthrough replication. Current data are not able to provide evidence that NMDTG protect from HIV transmission and cannot be interpreted as a proof of protection.*

However, for purpose of this manuscript, it is important to show in vivo efficacy of NMDTG, not necessary protection from HIV transmission. The experiment with PBL mouse model as well as the new experiment in figure 6 provide some evidence that the drug can be effective in vivo.

Response: Understood, Thank you. Amendments are made in the text for better clarification.

Point 3. *Please indicate in the paragraph describing the new experiment (lines 261-291) that the experimental design did not exclude possibility that high concentration of the drug present during analysis can suppress potential breakthrough infection.*

Response: We added the disclaimer in the text that the experimental design cannot exclude the possibility that high concentrations of drug present during analysis could suppress potential breakthrough of HIV-1 infection.

Point 4. *“Complete protection” is redundant term. One can be protected from HIV at the time of exposure or not. If HIV is detected in any time post-exposure, there was no protection. In lines 262,323, 331, and 333 this term is used for HIV analysis (concentration of RNA/DNA) below limit of detection of an analytical method. It should be described like that, not as a “complete protection”.*

Response: We changed the terminology “complete protection” to avoid any confusion and redundancy. We refer to our HIV analysis (concentration of RNA/DNA) as below the limit of detection.

Reviewer 2

Overall. *The major claims of this manuscript is myristoylation of DTG AND encapsulation of MDTG into poloxisamer to make NMDTG. NMDTG is easily taken up by MDM and is stable inside MDM and this made apparent half life of DTG very long. Only MDTG nor NDTG could not achieve this property. Previous manuscript was misunderstanding that MDTG is active. However, in the revised manuscript, hydrolysis of MDTG in biological fluid is very rapid and it was caused by esterases. Encapsulation*

of MDTG into polymer protects hydrolysis of MDTG. Other comments were well responded.

Response: Thank you.

Point 1. P14, L299, "physiochemical" may be "physicochemical".

Response: The word spelling was corrected as "physicochemical".

Point 2. My previous Point 3, I confirmed "reversible" was deleted. However, in page 4, L71, I noticed "reversible" still remains. If the authors intend this word for the meaning that eventually NMDTG is degraded and converted back to DTG, it is OK.

Response: We eliminated the term "reversible".

Reviewer 3

Overall. The re-submitted manuscript detailing the synthesis, characterization, and in vitro/in vivo evaluation of a lipid prodrug of Dolutegravir has gone a long way toward addressing nearly all of the comments and suggestions provided by reviewers in the first round. I commend the authors for the diligent and thorough follow-up. Additional data include:

- clearer details associated with API characterization (both graphical and written)
- additional pieces of in vitro data
- additional in vivo (PrEP) models to paint a clearer picture of DTG performance in mouse models despite caveats associated with these models.

These extensive additions/edits significantly strengthen the manuscript and indicate a true dedication of the author to his work. In conducting a final review, there remain some minor formatting/punctuation errors that should be corrected if possible. I have not captured these in writing, but they are primarily associated with the incorrect use of dashes (dash, n-dash, m-dash) throughout the manuscript. I would recommend the author review these in detail and correct appropriately.

Otherwise, thank you for re-submitting a well re-written manuscript. Congratulations on a very nice piece of work!

Response: Thank you very much. We corrected all observed formatting/punctuation errors and eliminated when possible any incorrect use of dashes.

Lastly, we thank the reviewer(s) for his/her kind words on the impact of our work.